# A standardized image processing and data quality platform for rodent fMRI

Gabriel Desrosiers-Grégoire [1,2] ✉, Gabriel A. Devenyi [1,3],
Joanes Grandjean[4,5] & M. Mallar Chakravarty [1,2,3,6] ✉

Functional magnetic resonance imaging in rodents holds great potential for advancing our understanding of brain networks. Unlike the human community, there remains no standardized resource in rodents for image processing, analysis and quality control, posing significant reproducibility limitations. Our software platform, Rodent Automated Bold Improvement of EPI Sequences, is a pipeline designed to address these limitations for preprocessing, quality control, and confound correction, along with best practices for reproducibility and transparency. We demonstrate the robustness of the preprocessing workflow by validating performance across multiple acquisition sites and both mouse and rat data. Building upon a thorough investigation into data quality metrics across acquisition sites, we introduce guidelines for the quality control of network analysis and offer recommendations for addressing issues. Taken together, this software platform will allow the emerging community to adopt reproducible practices and foster progress in translational neuroscience.

Functional magnetic resonance imaging (fMRI) experiments in rodents have enabled important investigations of brain activity organisation[1]. Several avenues are now available for probing the biological underpinnings of brain networks by coupling fMRI experiments conducted in model organisms with transcriptomics[2], calcium imaging (using wide-field[3] or fibre-photometry[4]), optogenetics[5,6], and more. Such applications hold great potential for bridging across biological and functional scales and, in turn, provide a powerful translational tool to address open questions pertaining to behaviour, brain development or neuropsychiatric disease.

The study of resting-state networks in rodents is a rapidly expanding neuroscientific discipline. Whereas workflow standardisation emerged for human fMRI applications[7,8], the rodent imaging community has yet to deliver agreed-upon methodological standards across various aspects of experimental designs. Recent multi-site studies have demonstrated significant variability in the ability to detect canonical brain networks in mice[9] and rats[10]. Accordingly, reproducibility concerns are becoming increasingly salient within the rodent

community, drawing parallels with the ongoing conversation within the human literature[11,12]. Despite recent successes towards the standardisation of rodent acquisition protocols[10], there are two unresolved problems: (1) laboratories primarily rely on custom-made solutions for image processing since no readily available software has been validated across multiple acquisition sites, and (2) there are no governing standards for data quality assessment for rodent images. The choices across image processing steps, including image alignment to a commonspace, head motion realignment, and appropriate confound correction, can have significant implications for downstream analysis outcomes; the impact of these experimental design choices further emphasises the need for a standardised state-of-the-art processing workflow in this domain[7]. Developing robust image processing workflows for rodent acquisitions presents new challenges due to site-specific field strengths (4.7, 7, 9.4, 11.7 T), coil types (room-temperature vs cryogenic) and geometry (i.e., surface vs saddle vs quadrature coil), and acquisition parameters[9,12]. Although rodent-adapted software pipelines were recently introduced for either mice and rats[12–16], no

¹Computational Brain Anatomy Laboratory, Cerebral Imaging Center, Douglas Mental Health University Institute, Montreal, QC, Canada. ²Integrated Program in Neuroscience, McGill University, Montreal, QC, Canada. ³Department of Psychiatry, McGill University, Montreal, QC, Canada. ⁴Donders Institute for Brain, Cognition and Behaviour, Radboud University Medical Center, Nijmegen, The Netherlands. ⁵Department of Medical Imaging, Radboud University Medical Center, Nijmegen, The Netherlands. ⁶Department of Biological and Biomedical Engineering, McGill University, Montreal, QC, Canada. ✉e-mail: gabriel.desrosiers-gregoire@mail.mcgill.ca; mallar.chakravarty@mcgill.ca

existing software has been thoroughly validated across acquisition sites and species in terms of preprocessing quality. In addition, thorough data quality assessment is also essential for comparability between studies. In the human literature, after noting that poor accounting for confounds may lead to false positive results[17], there has been significant debate regarding how to appropriately account for fMRI confounds during connectivity analyses. The human fMRI community recently came together to delineate general insights and recommendations that can be drawn from the quality control protocols of several research teams[18]. In rodents, investigations into confound characterisation are emerging[15,19], but in spite of the aforementioned acquisition protocol guidelines[10], sources of divergence in data quality between datasets remain, and rodent-adapted guidelines for quality control are still needed for the community.

In this work, we aim to address the need for standardised workflow and quality control measures in network analyses. We address these shortcomings with two main innovations: (1) the implementation of an adaptive image registration workflow offering consistent, high-quality outputs across sites and species, and (2) conducting a thorough inspection of data quality markers across acquisition protocols to introduce reliable guidelines for quality control in network analysis. Inspired by previous initiatives fostering reproducibility and standardisation in humans[7,18,20], these innovations are integrated through the distribution of an open-source software platform, Rodent Automated Bold Improvement of EPI Sequences (RABIES; publicly available to the community https://github.com/CoBrALab/RABIES and thoroughly documented online https://rabies.readthedocs.io/en/stable/) that provides an integrative solution for preprocessing, confound correction, analysis, together with data diagnostic tools encouraging best practices for quality control. While the RABIES pipeline was used elsewhere for preprocessing in rats[10], in the current manuscript, we formally describe the software's technical implementation and its validation across rodent species and develop a suite of data quality reports and guidelines tailored for tackling comparability between rodent datasets.

## Results

### Software designed to facilitate reproducible science and transparency

RABIES is an open-source software largely inspired by fMRIPrep's design for reproducibility[7]. We made three important design choices that follow best practices for reproducibility in neuroimaging research: (1) the input data format requires that users follow the BIDS standards[21]; (2) quality control can be conducted using automatically generated reports; and (3) the software distribution is handled via containerised versions (https://github.com/CoBrALab/RABIES/pkgs/container/rabies; executable through Docker or Singularity[22]), ensuring reproducible execution across computing environments. Like fMRIprep, RABIES is built using Nipype[23], which allows optimal resource allocation for parallelisation and for a flexible workflow reorganisation according to provided input parameters and dataset features (e.g., number of subjects, the availability of structural images, and the presence of multiple fMRI sessions). Log and quality control reports are automatically generated in portable, standardised formats, allowing for rapid and reliable consultation while encouraging sharing along presentations and publications.

The major pipeline steps include a registration workflow conducting essential preprocessing steps prior to group-level analysis (Fig. 1a and Supplementary Table 1), the quality control of registration operations using a visual report (Fig. 1b), confound correction (Fig. 1c), and connectivity measurement in individual scans (Fig. 1d), altogether providing outputs prepared for subsequent statistical analyses and hypothesis testing. Outputs from each pipeline stage are generated in standardised NifTi or CSV formats, readily available for exportation and subsequent uses external to RABIES.

### A robust registration workflow across rodent species and imaging sites

To evaluate the robustness of our registration workflow, we used multi-site fMRI data spanning field strengths (4.7, 7, 9.4, and 11.7 T), coil types (room temperature and cryogenically cooled coils), anaesthesia protocols (awake, isoflurane [iso], medetomidine [med], med and iso, or halothane), ventilation protocols (free breathing and ventilated), and rodent species (mice and rats) (23 publicly available datasets total, 441 mice and 232 rats echo-planar imaging (EPI) scans, 29 EPI scans per dataset on average; Supplementary Table 2). The visual quality control report (Fig. 1b) was examined across datasets to benchmark every critical registration operation throughout the pipeline, consisting of brain masking for B1 inhomogeneity correction, alignment to commonspace and EPI susceptibility distortion correction (methods section 4; Supplementary Fig. 1). Several innovations were required to ensure high-quality image alignment in all 23 datasets (detailed in Supplementary Table 3), given the extensive variability in species, inhomogeneity artefacts, brain coverage and anatomical contrast of the EPI images (Supplementary Fig. 2). Despite a common workflow, mild differences in preprocessing parameters per dataset were necessary to achieve ideal performance across all datasets (listed in Supplementary Table 4).

Each dataset was processed individually using the final workflow, and performance was evaluated by inspecting the quality control report, where a 'successful' registration would provide sufficient image alignment across the brain for most standard analyses (see details in methods section 4). In the improved workflow, failures were marginal (Supplementary Table 2 and Supplementary Fig. 3). For the 17 mouse datasets lacking structural images, we achieved a 100% (255/255) success rate for the B1 intensity inhomogeneity correction step in EPIs and 99.6% (254/255) success during cross-subject alignment to the unbiased template. For the 3 mouse datasets, including supporting structural scans, we obtained a 100% success rate for intensity inhomogeneity correction in both structural (110/110) and EPI (185/185) scans. We achieved a success rate of 99.1% (109/110) for the alignment to the unbiased template and 100% (185/185) for the susceptibility distortion correction of EPIs through non-linear alignment to the structural scan. Finally, in the 3 rat datasets (1 dataset only included EPI scans), 98.9% (88/89) and 100% (230/230) of scans passed inhomogeneity correction for anatomical and EPI scans, respectively, and we obtained 99.6% (230/231) success rate for cross-subject alignment and 98.9% (87/88) success rate during susceptibility distortion correction. Detailed reports for each dataset can be found in Supplementary Table 2, and the visual quality control report from each preprocessing run is provided in the Supplementary files. We consider that this high success rate can address the significant preprocessing needs within the rodent fMRI community, despite the current cross-site variability in acquisition equipment and parameters.

### Standard tools for confound correction and connectivity analysis

Following image preprocessing, RABIES integrates a selection of tools for conducting confound correction (Fig. 1c and Methods section 5) and for deriving connectivity measures (Fig. 1d and Methods section 8). The available options for confound correction[24] cover most standard strategies in the literature, including censoring (or scrubbing)[17], confound regression[25], and frequency filtering[24]. Operations are combined and orchestrated to maximise effectiveness while preventing the re-introduction of artefacts (e.g., ringing artefact when filtering prior to censoring[26]), as recommended in Power et al.[27] and Lindquist et al.[28]. Following confound correction, connectivity can be measured in each individual scan, and results then exported for statistical testing. Analysis options include seed-based correlation[29], whole-brain functional connectivity matrix generation[30], and dual regression[31,32]. Integrating

standardised operations for confound correction and connectivity facilitates the reproducibility of these steps as a means of improving comparisons across studies[33].

## Rodent-adapted guidelines for data quality assessment

Defining interpretable and reliable quality control measures is challenging. Most common practices in humans include estimating confounding effects from non-neural sources (e.g., motion or respiration) on connectivity measures[8]. In addition in rodents, canonical network detectability varies significantly between acquisition sites both in mice[9] and rats[10] irrespective of non-neural confounds. This could be explained by the relatively lower signal-to-noise ratio (SNR) in small animals and/or anaesthesia effects (known to decrease network activity)[1,34]. Canonical network detectability is essential for experimental reproducibility and should be accounted for during quality control. In the remainder of the results, we expand on previous work in rodents (Supplementary Discussion) to define guidelines for network analysis quality control across two separate axes: spurious effects from confounds and network detectability. We identify metrics that capture key features of spurious connectivity and network detectability at the

scan level (i.e., scan diagnosis), from which we derive guidelines to conduct quality control prior to group statistics. Finally, we discuss how these tools can support the improvement of data quality through optimisation of confound correction, as well as the revision of analysis designs and acquisition protocols. The tools described in this section follow upon previous stages of the RABIES pipeline, leading to the generation of connectivity outputs (Fig. 1), allowing evaluation of confound correction and analysis design as well as informing subsequent statistical testing.

## Data quality markers identified through scan diagnosis

To define an appropriate set of metrics benchmarking data quality, a large set of features was assessed from the fMRI scans across multi-site mouse datasets covering a comprehensive range of quality ($N = 19$ datasets, 17 sites, 367/369 scans passing preprocessing quality control; see Methods section 9). This was conducted using a scan-level qualitative visual report (i.e., spatiotemporal diagnosis; Supplementary Methods) which compiles metrics featuring temporal and spatial properties of the blood-oxygen-level-dependent (BOLD) signal. From these observations, we determined that scan quality can be

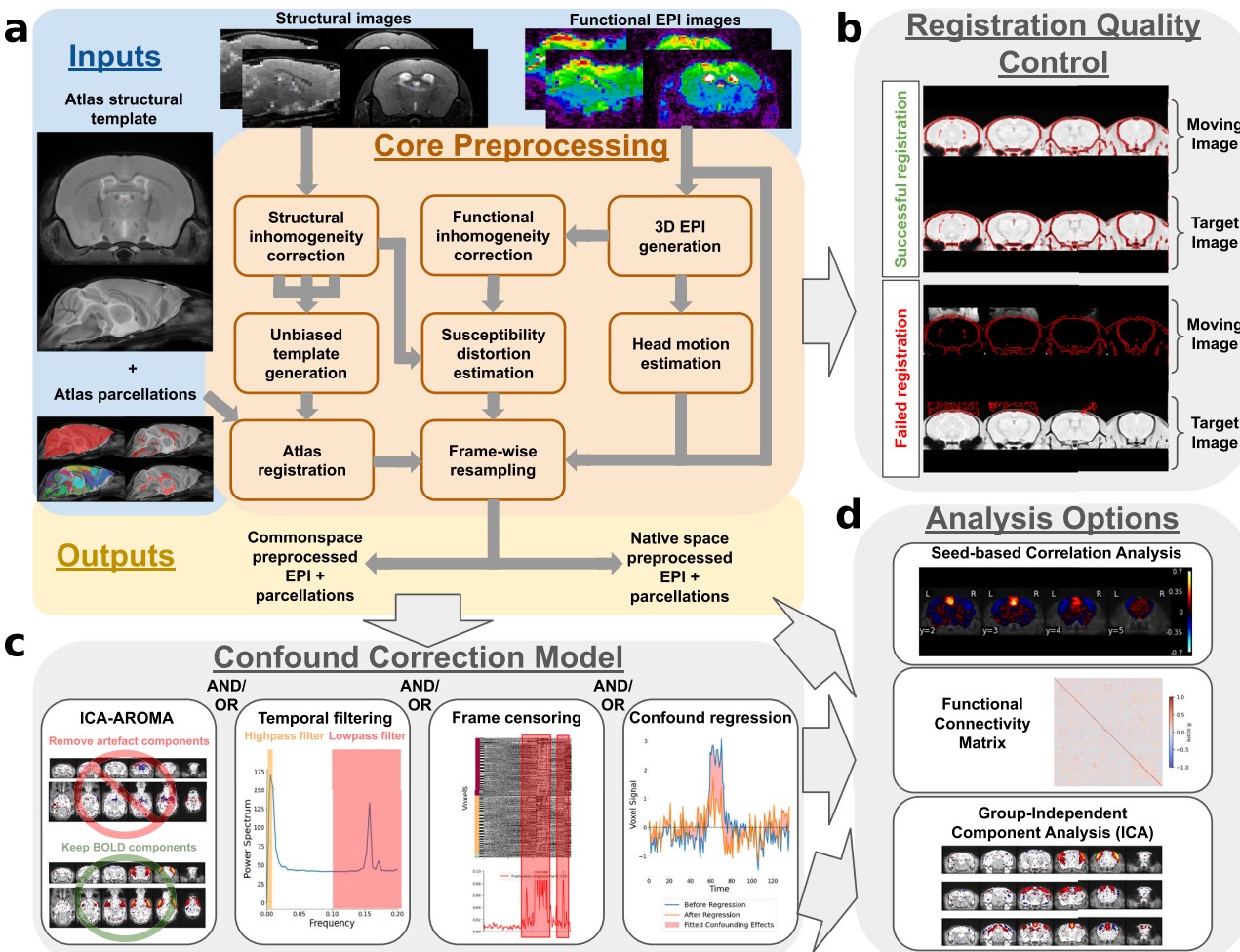

**Fig. 1 | RABIES allows the preprocessing, quality control, confound correction and analysis of rodent fMRI datasets. a** Schematic organisation of the preprocessing pipeline. RABIES takes as inputs a dataset of EPI functional scans along with their structural scan (optional) in BIDS format, together with an external reference structural atlas containing anatomical masks and labels. The core preprocessing architecture conducts commonspace alignment, head motion realignment and susceptibility distortion correction (detailed in methods section 1; sup. table 1). **b** Along the execution of the preprocessing pipeline, RABIES generates

automatically PNG images, allowing the visual quality control of each failure-prone preprocessing step (methods section 4). **c** Following preprocessing, an array of confound correction strategies are made optional and can be customised according to user needs (methods section 5). **d** After applying confound correction, a final workflow is made available to conduct basic resting state analysis (methods section 8). EPI: echo-planar imaging; RABIES: Rodent Automated Bold Improvement of EPI Sequences.

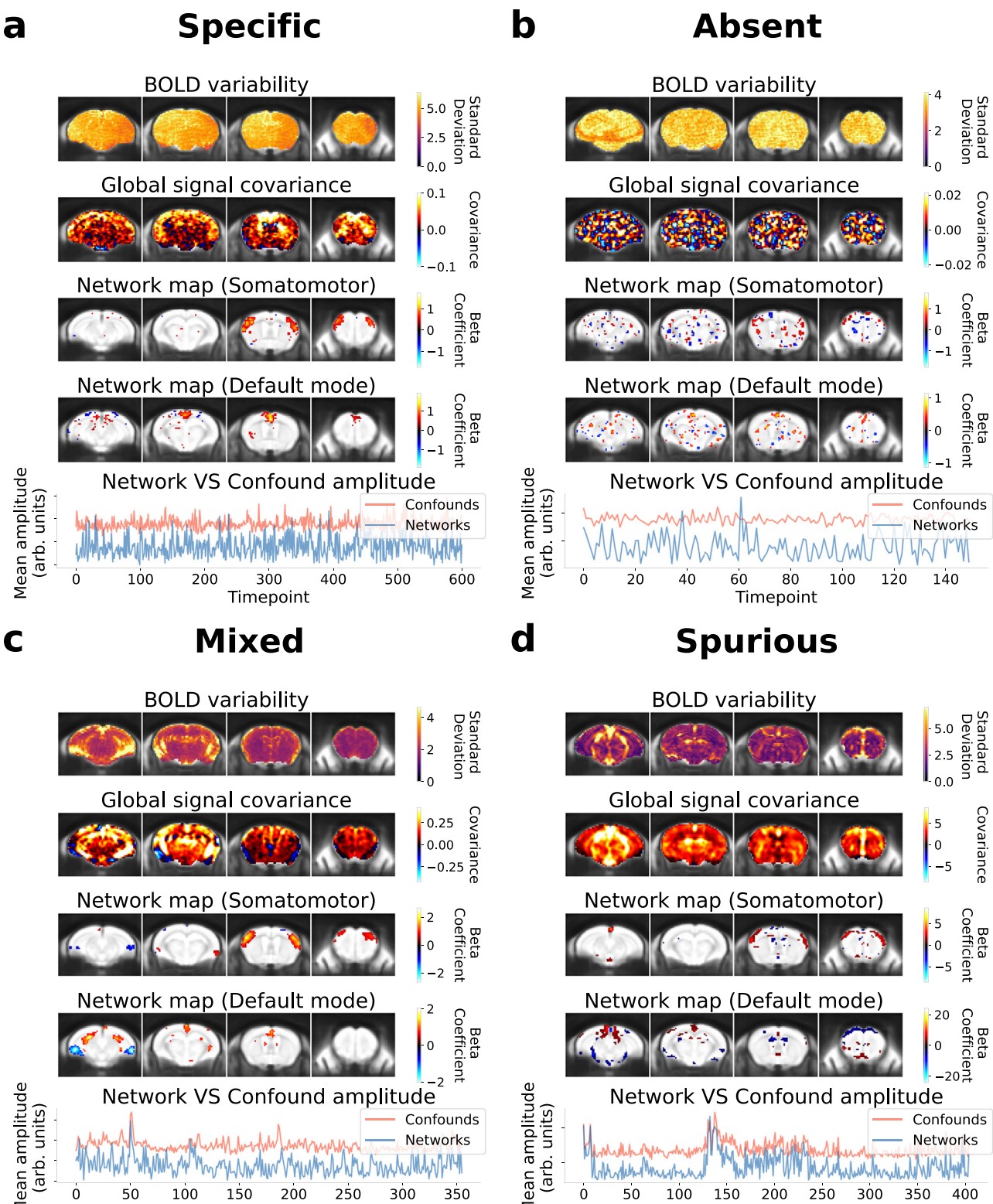

categorised into four main groups (Fig. 2): specific connectivity (i.e., the network is well detected) without confound signatures (specific), confounds leading to spurious connectivity (spurious), no network connectivity nor confound signatures (absent), or scans which did not neatly fit within one category, with a mixture of network and confound signatures (mixed). A subset of features from the spatiotemporal diagnosis are key in characterising these quality divergences, and are detailed below:

A. BOLD variability: The temporal standard deviation at each voxel reflects the spatial distribution of signal variability. The resulting BOLD variability map presents a homogeneous contrast in uncorrupted scans and can otherwise reveal the anatomical signature of confounds, thus allowing us to identify the type of confound (Supplementary Figs. 4, 5).

B. Global signal covariance: The global signal covariance map (the temporal covariance of each voxel with the global signal

**Fig. 2 | Four main categories of scan quality.** An example scan is displayed for each category using a key subset of features from the spatiotemporal diagnosis (Supplementary Methods; Supplementary Fig. 4 and Supplementary Table 5). Network maps were derived through dual regression in these examples and are thresholded to keep the top 4% of voxels with the highest absolute value. Each anatomical map is displayed across four coronal slices overlapped onto the commonspace reference template. In the last panel, timecourses belonging to the network or confound category are derived from dual regression, and a representative average is displayed for each category. To do so, each time course (from a network or confound) is initially variance normalised with an L2-norm and an average is derived by taking the mean across the absolute-valued (i.e., positively-weighted) time courses within a given category. **a** In the specific category, the BOLD variability is mostly homogeneous, the contrast of the global signal covariance is predominantly in the grey matter without spurious features of confounds, the network maps reproduce features of the canonical network, and the networks do not display obvious correlation with confounds across time. **b** In the absent category, the contrast between the global signal covariance and network maps is largely random. **c** In this example scan showing mixed features, spurious signatures are observable along anatomical edges in the BOLD variability map and overlap with features of the global signal covariance. Core network features are derived from the network maps, although there are also some spurious features. Some correlations can be noticed between the network and confounded timecourses. **d** In the spurious category, signatures of confounds predominate in all features. In this example, a confound signature is predominant in the ventricles and major brain vessels, as observed from the BOLD variability and global signal covariance. The network maps resemble spurious features, and the network time course is predominantly correlated with confounds. BOLD: blood-oxygen-level-dependent signal.

timecourse), is sensitive to coordinated fluctuations across voxels (i.e., does not include random noise as with the BOLD variability). As such, it is sensitive to both non-neural confounds (e.g., Fig. 2d) and network signatures (e.g., Fig. 2a). The global signal covariance thus reflects whether network or confound sources dominate coordinated fluctuations, and can delineate the most likely contributors to downstream connectivity measures.

C. Network map: When conducting dual regression or seed-based connectivity analysis, the network connectivity map is displayed for each individual network to determine if regions known to belong to specific networks are effectively captured by the connectivity analysis (i.e., network specificity). As reported in the original publication of the multi-site datasets[9], in certain scans, the network was absent, or the network shape was distorted and shows spurious connectivity signatures.

D. Network and confound timecourses: Finally, the respective timecourses for networks and confounds can be compared to reveal direct relationships between network amplitude and confounds in the temporal domain. Although this metric does not describe the type of confound, it is the most direct indicator of spurious connectivity. It is an important complement to the inspection of network shape, since spurious effects may only affect amplitude with minimal impact on shape (Supplementary Fig. 6). To model confounded timecourses, dual regression analysis is conducted with a complete set of components from Independent Component Analysis representing a mixture of networks and confounds from various origins (Methods section 10), and the timecourses from confound components are compiled to summarise a broad set of potential confounds.

These 4 features robustly capture the essential characteristics of network detectability and spurious connectivity at the single scan level. The remaining features from the spatiotemporal diagnosis provide additional details regarding timeseries properties, the motion parameters, or confound regression, and can further support confound characterisation (Supplementary Table 5). Altogether, this in-depth investigation at the scan level allows for detailed reporting of quality features in a given dataset, and supports defining key dimensions of data quality to control for during analysis.

## A quality control framework for network analysis

While the scan-level observations described above allow for characterising specific data quality pitfalls, it does not establish how these may impact group-level statistical inference. Statistics in network analyses are commonly controlled for using group-level quality indices relating connectivity to confound metrics. For instance, the group-level correlation between mean framewise displacement and connectivity can be estimated across subjects[8]. Here, we provide recommendations in the form of systematic guidelines, to control for both network detectability and spurious connectivity issues prior to statistical analysis. To avoid absent and spurious connectivity (Fig. 2),

individual scans that do not meet minimum quality thresholds are first removed. Subsequently, group-level statistical reports describing the impact of data quality on inter-scan variability in connectivity are generated. To allow for the inspection of network shape (see below) and to identify network-specific issues, these guidelines are carried out a per network basis. Supplementary Fig. 4–12 document observations derived using RABIES reports across the aforementioned mouse datasets to support these guidelines.

Scans which fail on either standard of network detectability or spurious connectivity are removed using two scan-level measures (Fig. 3):

- Network specificity: The specificity of the network map relative to the canonical network is evaluated using the Dice overlap of the thresholded maps. Sufficient specificity implies both network detectability (i.e., see the absent category in Fig. 2b) and minimal spurious effects distorting network shape (see spurious scan category in Fig. 2d).
- Confound temporal correlation: To evaluate whether the network timecourse is dependent on known confounds (indicating spurious connectivity), the timecourse is correlated with confound sources evaluated from dual regression (after the careful identification of confound components described in methods section 10). This measure complements the Dice overlap by accounting for spurious effects on network amplitude and allows us to fully distinguish the spurious and absent categories.

We propose that scans which do not meet thresholds on either measure be removed from downstream statistical analysis. These thresholds should be selected by inspecting quality markers from Fig. 2, identifying scans with spurious or absent connectivity, and determining an appropriate threshold for each metric to delineate these scans. In addition to these two thresholds, for dual regression analysis, scans that present outlier values in network amplitude are removed (Methods section 14). This conservative approach was taken as extreme values of network amplitude may not be biologically plausible and can present additional evidence of spurious effects[32].

After removing individual scans, network detectability and spurious effects are evaluated at the group level based on inter-scan variability using the following metrics compiled in a dataset statistical report (Fig. 4a):

- Specificity of network variability: The inter-scan standard deviation in connectivity is computed voxelwise, creating a network variability map which is then compared to the canonical network map. In several datasets composed of scans with specific connectivity, we observed that network variability is largely specific to the anatomical extent of the network modelled. Conversely, datasets characterised by absent connectivity present random variability distributions, and heavily confounded datasets display spurious connectivity signatures (Fig. 4b). Evaluating the

specificity of network variability thus allows to mitigate issues of absent or spurious connectivity which may persist at the group level despite scan-level quality control (Supplementary Figs. 9, 12). The expectation of network variability for quality control is further discussed in the Supplementary Discussion.

- Confound effects: Connectivity is correlated across scans with three complementary measures of confounds: mean framewise displacement, the variance explained from confound regression, and temporal degrees of freedom (tDOF). Mean framewise displacement is amongst the most commonly used metrics[8]. However, it is limited to motion effects and does not localise confounding effects anatomically. Conversely, the variance modelled from confound regression, computed voxelwise, can reliably capture a wider variety of confound sources with anatomical specificity (Supplementary Figs. 4, 5). Lastly, unequal tDOF between different scans, resulting, for instance, from frame censoring during confound correction, can introduce further statistical biases into the evaluation of connectivity strength. To mitigate confound effects, correlations should be minimised for all three metrics.

Scan-level thresholds must be applied together with the evaluation of group-level statistics as group-level metrics are inaccurate without meeting data quality assumptions at the scan level, and group statistics inspecting inter-scan variability can reveal persisting issues pertaining to group analysis despite sensible scan-level features (Supplementary Table 6). However, each quality control metric has limitations (Supplementary Table 6), and certain issues are not always best captured through a simplified framework. Ideally, the interpretation of the quality control report should be supported by the inspection of the spatiotemporal diagnosis in individual scans, which provides a more comprehensive evaluation of dataset-specific issues.

**Applications for improving analysis quality markers**

Here we describe potential applications of these tools for improving data quality and orienting analytical design. We first explore how confound correction can be optimised according to the relevant quality issues, then discuss further applications for defining analysis design and improving acquisition protocols.

Ideally, the confound correction strategy is adapted to correct the specific issues present in a given dataset, given that excess correction can negatively impact data quality by removing network activity[35]. This was tested by comparing the impact of various correction methods available within RABIES on the scan-level and group-level quality. We observed important trade-offs across different aspects of data quality (e.g., reduction of confound correlation at the cost of network detectability), and important variability between datasets regarding the effectiveness of a given correction strategy (Supplementary Fig. 13). Thus, instead of applying the same strategy across datasets, we leveraged the spatiotemporal diagnosis and analysis quality control to define a principled approach to designing dataset-specific corrections which address relevant aspects of data quality (see Methods section 15, Supplementary Fig. 14, and the rationale discussed in Supplementary Discussion). With this approach, correction strategies are combined incrementally, evaluating scan quality features and analysis quality control at each iteration, with the goal of maximising network detectability and minimising spurious effects. This process was conducted across the multi-site datasets and led to improvements in quality, both at the scan (36 more scans passing threshold for confound effects, 7 less passing for network specificity) and dataset level (3 more datasets passing threshold for network specificity and 2 more for confound effects) (Fig. 5). In particular, correlations with confounds were greatly reduced at the scan level, and network specificity was improved in datasets with spurious connectivity, suggesting

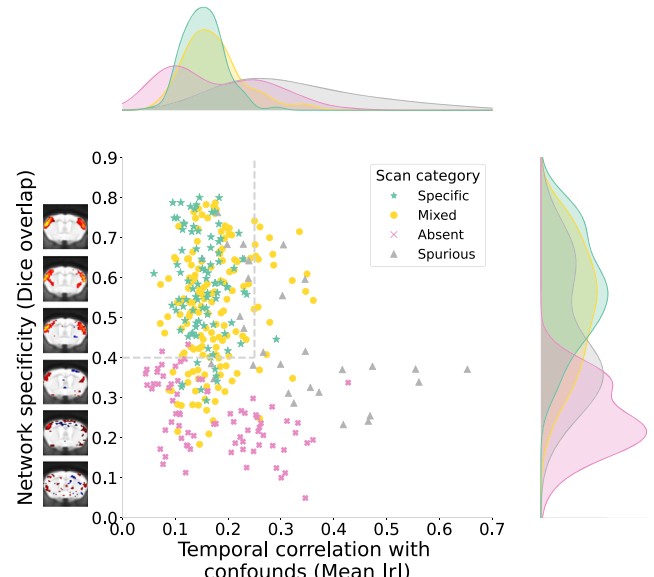

**Fig. 3 | Scan-level network specificity and confound correlation across 19 datasets.** The results shown correspond to the analysis of the somatomotor network with dual regression. Each dataset was assigned a predominant scan category (Supplementary Table 2), and the scans were labelled accordingly. Network specificity is estimated by first thresholding the network map to identify the top 4% of voxels with the highest connectivity, and then a Dice overlap is computed between the area covered by those voxels and the area covered by the corresponding canonical network map (Methods section 12) thresholded in the same way. Both scan and canonical network maps are spatially smoothed with 0.3 mm FWHM before thresholding. The temporal correlation with confounds is estimated as the mean absolute correlation between the time course of the network and the time course from each confound component obtained through dual regression. The grey dotted lines represent the selected thresholds for inclusion. The threshold for network specificity (Dice> 0.4) is selected to delineate scans which reproduce network shape with little spurious features. A set of examples is shown along the axis of network specificity to represent corresponding increments in Dice overlap. For the confound temporal correlation, a threshold of 0.25 is selected to delineate scans in which clear spurious features were observed from the quality markers in Fig. 2.

correction of spurious effects on network shape. However, in datasets with low network detectability (i.e., absent connectivity), confound correction was unable to improve quality, suggesting a limited network signal at data acquisition instead, which cannot be recovered by correcting confounds. Thus, this protocol offers important support for navigating the range of options for confound correction and mitigating spurious effects on connectivity, but alternatives must be considered for issues strictly targeting network detectability.

Other important aspects of the experimental design which impact network quality are the analysis design and the acquisition protocol. The impact of data quality issues is influenced both by the technique selected for measuring connectivity and the specific brain network studied (Supplementary Figs. 15, 16). We observed greater challenges with regard to network detectability with the default mode network than the somatomotor network, which could be explained by an increased susceptibility to physiological confounds provided its overlap with major blood vessels and ventricles (Supplementary Fig. 8), a narrower anatomical shape compared to the somatomotor network, an increased anaesthesia susceptibility, or a reduced intrinsic activity of the network. In addition, although seed-based connectivity does not require defining network maps a priori as with dual regression, this technique performed relatively worse on measures of network specificity. Thus, the network of interest and type of analysis may be best considered in the context of data quality, supported by

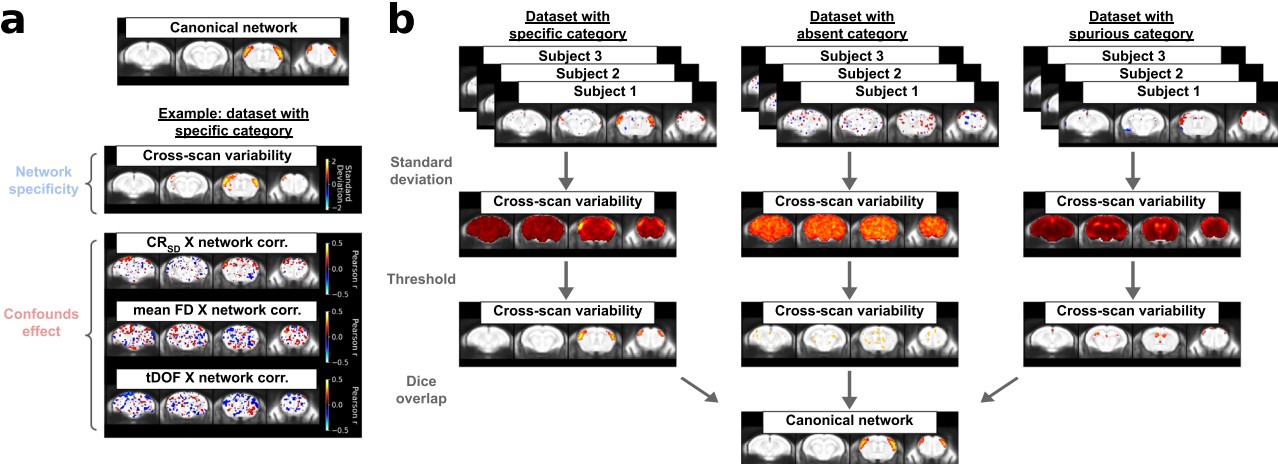

**Fig. 4 | The group-level statistical report for quality control of network analysis. a** An example is shown for a dataset which displayed desired outcomes for the somatomotor network using dual regression. The cross-scan variability is measured with the standard deviation at each voxel, and the resulting brain map is thresholded to include the top 4% of voxels with the highest value. The predominant variability is specific to the canonical network (shown at the top, thresholded the same way). The correlation maps (thresholded at $r > 0.1$) for each confound measure are then listed. For mean FD and tDOF, a single scalar is derived for each scan, and correlated with connectivity across subjects for each voxel. Since $CR_{SD}$ is measured at each voxel (Supplementary Fig. 4), this measure is correlated voxelwise with connectivity, allowing to delineate effects which are anatomically specific (Supplementary Fig. 8). All brain maps in the report are spatially smoothed with a 0.3 mm FWHM kernel before display and thresholding. **b** Schema illustrating the evaluation of the specificity of network variability and example cases for the specific, absent and spurious categories. After deriving thresholded maps of network variability and the canonical network, specificity can be summarised quantitatively by estimating the Dice overlap between the thresholded regions. $CR_{SD}$: variance explained by confound regression (Supplementary Fig. 4); mean FD: mean framewise displacement; tDOF: temporal degrees of freedom. Source data is provided as a sheet in the Source Data file.

network quality control guidelines. Finally, the RABIES tools can support accurately identifying underlying issues at acquisition (e.g., significant motion, physiological instability or low network detectability), and in turn, these observations may help users in designing adjustments to the acquisition protocol (e.g., improving head stability, increasing or lowering anaesthesia administration) or other aspects of the experimental design. In cases where networks are detected at the scan level, but network variability is absent at the group level, more scans may need to be acquired, as this metric is dependent on sample size (Supplementary Fig. 12b).

Altogether, data quality assessment within RABIES supports several aspects of the analysis workflow. For best practices, these tools can be integrated throughout an experiment as it unfolds, orienting researchers in designing high-quality experiments, and ensuring quality standards are met for the final analyses (Fig. 6).

## Discussion

We introduce a robust image processing and analysis pipeline adapted for specific challenges to rodent fMRI. While previous preprocessing pipelines were introduced for rodent fMRI[12-16], the strengths of the proposed approach include a thorough validation across acquisition sites and species, spanning broad differences in image quality, together with the integration of state-of-the-art tools for confound correction, analysis and data quality assessment. To date, RABIES is also the only rodent software combining containerised distribution, BIDS compatibility and extensive user documentation. These advances with regard to accessibility are essential for the widespread adaptation of standard practices across the rodent imaging community.

Motivated by the current variability in results from network analysis between mouse fMRI sites[9,10], we provide insights into the underlying sources of data quality divergences leading to downstream issues with network analyses. We put forward tools and guidelines for the quality control of network analysis, which support addressing potential data quality challenges and deriving informed decisions regarding experimental design. Establishing formal guidelines in those regards is challenging, and remains an ongoing topic of conversation in the human literature[18]. Hence, our approach leverages a wide array of tools for quality control (Supplementary Table 5) and synthesises recommendations based on observations derived across a representative variety of rodent fMRI acquisition protocols. This is an important initial step towards providing standards for high quality rodent connectivity research, and their adoption by the wider community will improve comparability between studies, as well as foster future conversations towards addressing persisting challenges.

Despite the innovations of the RABIES software, certain limitations remain with regard to reproducibility within the current workflow. To address the substantial variability in image contrast and artefacts across rodent acquisition protocols, it remains necessary to provide flexible registration parameters which must be handled by the user. We provide a set of recommendations online to navigate the variety of registration challenges (https://rabies.readthedocs.io/en/stable/registration_troubleshoot.html). Also, the quality control of registration relies on subjective visual inspection, although this limitation is shared with human work[7,20]. There are further concerns surrounding the design of confound correction, which could not generalise across datasets, drawing parallels with ongoing challenges discussed in the human literature[36,37]. Here, we put forward a protocol which, informed by the reports from analysis quality control, allows identifying the most sensible strategies for correcting issues pertaining to a given dataset. Thus, instead of putting forward a single solution for confound correction, we support the dataset-specific design and recommend harmonising standards along with quality control measures accounting for issues of network detectability and spurious connectivity. Similar to the quality control of preprocessing, the proposed guidelines for the quality control of network analysis rely on the visual interpretation of reports comprising several metrics. This is necessary, as generic quality control metrics have their shortcomings (Supplementary Table 6), and their affiliated thresholds may not blindly generalise across all datasets. Instead, as highlighted in a recent editorial, the bedrock of quality control with fMRI still rests on qualitative assessment of the data, which can then support quantitative methods[18]. For this purpose, RABIES provides

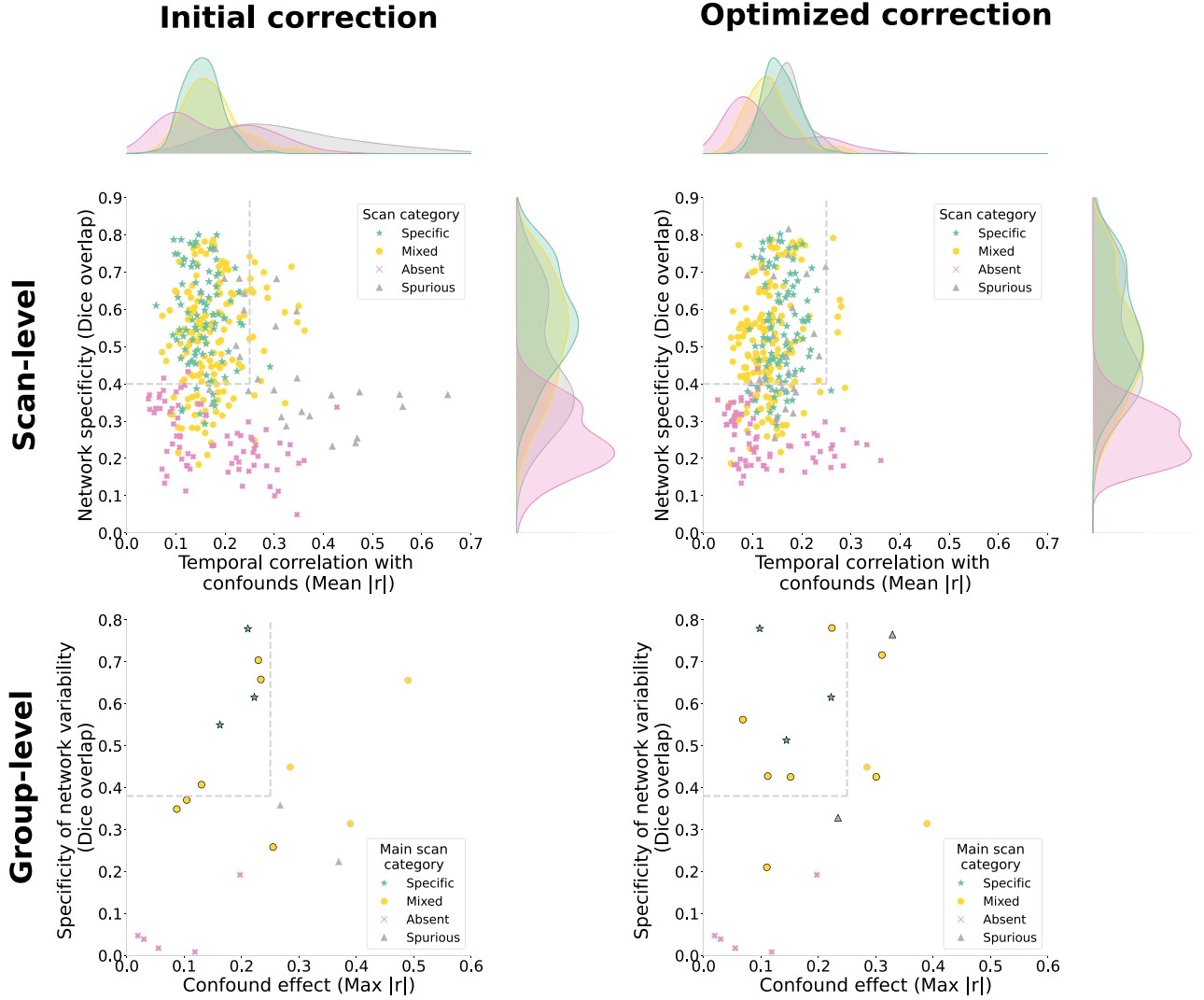

**Fig. 5 | Impact of optimising confound correction on quality control measures.**
Results are shown for analyses of the somatomotor network measured with dual
regression. At the top, scan-level measures are shown in Fig. 3 before and after
optimisation. With optimisation, the confound correlation is broadly reduced,
whereas network specificity is only significantly improved in the spurious category.
At the bottom, outcomes for group-level measures are summarised across all 19
datasets considered (each point is a dataset). The specificity of network variability
is estimated using Dice overlap with the canonical network map as illustrated in
Fig. 4b. For the confound effects, the mean correlation is calculated within the
thresholded area of the canonical network map for each of the three confound
metrics, and the maximum absolute mean correlation out of the three metrics is
shown (we take the absolute value to include negative relationships). Similarly to
the scan-level thresholds, the grey dotted lines represent ideal outcomes. The
threshold for network variability (Dice > 0.38) was determined upon visual

inspection of the spatial maps to attempt to distinguish maps displaying spurious
features. For the confound effects, a threshold of 0.25 was selected. Limitations to
these thresholds are discussed in the Supplementary Discussion. Points are circled
in the black when at least 8 scans pass scan-level thresholds before computing
group-level measures (points are circled in grey otherwise, and group-level mea-
sures are computed on all scans without applying scan-level thresholds). With the
optimisation of confound correction, the number of datasets passing this inclusion
threshold of 8 scans increased from 9 to 12. In the absent category, we can observe
that none of the 5 datasets could pass this threshold with or without optimisation.
The optimisation increased the specificity of network variability in several datasets,
where only three datasets that do not belong to the absent category do not meet
the threshold. However, confound effects remain above the threshold in several
datasets. Source data is provided as a sheet in the Source Data file.

adequate automatically-generated visual reports to complement the
interpretation of quality control metrics and the selection of appro-
priate thresholds. Finally, although the quality control guidelines we
present here can greatly support evaluating the main pitfalls in net-
work analysis, these recommendations are not meant to be required
irrespective of the scientific question and experimental design. The
judgement of the experimenter remains paramount in determining
which aspect applies to their study (for instance, network detectability
may not always be expected, if studying the impact of anaesthesia or
inspecting a visual network in blind subjects). The guidelines are thus
meant to offer a baseline which may apply to most standard resting-
state network studies, as well as to instruct new researchers and

RABIES users as to the main confounding factors which may impact
analysis.

We provide the community with a reliable tool for conducting
image processing together with recommendations for analysis quality
control, thereby allowing harmonisation of computational practices
across laboratories and fostering reproducible research. This
advancement is timely in light of current inconsistencies in metho-
dological and reporting practices for rodent fMRI. Like previous neu-
roimaging softwares[7,8,20,38,39], RABIES is made open source to
encourage community involvement in the development of additional
features (see our online guidelines https://rabies.readthedocs.io/en/
stable/contributing.html). This will support expanding beyond the

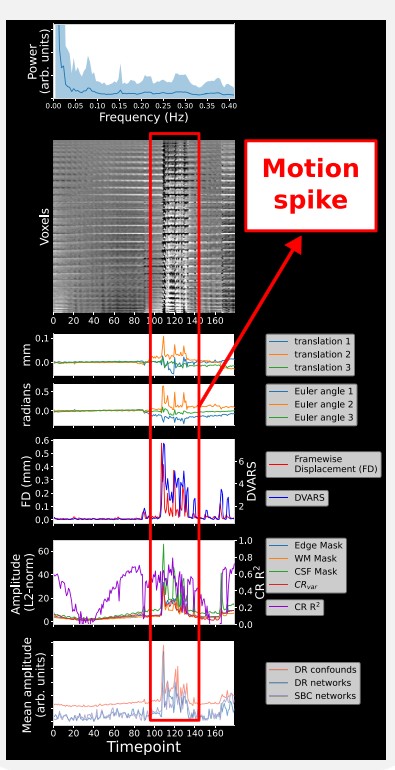

**Fig. 6 | Application of the RABIES data quality assessment tools.** Quality control criteria are first assessed to meet the recommended standards for analysis. These consist of three complementary steps, namely the visual assessment of scan-level markers for specific connectivity, consulting the dataset distribution plot and associated files to evaluate scan-level quality thresholds (Methods section 13), and consulting the group statistical report to evaluate expectations of network variability and group-wise confound correlation. If issues are found, these can be further characterised using the extended features from the spatiotemporal diagnosis (Supplementary Methods), together with the aforementioned quality control reports, to identify plausible sources for the issue. Here, the temporal features from a scan presenting a motion spike are shown. Finally, the information provided can be leveraged to improve experimental design, either by selecting an appropriate confound correction strategy (Supplementary Fig. 14), by revising the analysis approach, or by re-visiting the acquisition protocol. Following modifications, the quality control criteria can be re-assessed to evaluate improvements, and whether desired outcomes are achieved.

default workflow presented here to address more specific needs (e.g., inclusion of alternative distortion correction methods or multi-echo preprocessing) and, ultimately, involve the community in meeting evolving best practices over time. Previous initiatives, such as fMRI-Prep, have already been transformative in those regards for human work. The RABIES analysis quality control framework will be crucial for the development of high-quality data acquisition protocols[10]. Providing these reports, automatically generated from RABIES, along with publications, will improve transparency of results and prevent false positives/negatives. The adoption of RABIES for rodent fMRI will foster key practices and provide essential tools for addressing pressing issues in translational neuroscience.

## Methods

### Core preprocessing pipeline

The preprocessing of functional magnetic resonance imaging (fMRI) scans prior to analysis consists of, at minimum, the anatomical alignment of scans to a common space, head realignment to correct for motion, and the correction of susceptibility distortions arising from the echo-planar imaging (EPI) acquisition of functional scans. The core preprocessing pipeline in Rodent Automated Bold Improvement of EPI Sequences (RABIES) carries each of these steps with state-of-the-art processing tools and techniques (detail of each step in Supplementary Table 1).

To conduct common space alignment, structural images, which should be acquired along the EPI scans, are initially corrected for inhomogeneities and then registered together to allow the alignment of different MRI acquisitions. This registration is conducted by generating an unbiased data-driven template through the iterative linear and non-linear registration of each image to the dataset consensus average (using ANTs' SyN registration algorithm[40] with Mattes Mutual Information and ANTs cross-correlation objective functions for linear and non-linear stages, respectively), where the average gets updated at each iteration to provide an increasingly representative dataset template (https://github.com/CoBrALab/optimized_antsMultivariateTemplateConstruction)[41]. The finalised template after the last iteration provides a representative alignment of each MRI session to a template that shares the acquisition properties of the dataset (e.g., brain shape, field of view, anatomical contrast,...), making it a stable registration target for cross-subject alignment. This newly-generated unbiased template is then itself registered to an external reference atlas to provide both an anatomical segmentation and a common space comparable across studies defined from the provided reference atlas.

The remaining preprocessing involves the EPI images. A volumetric EPI image is first derived using a trimmed mean across the EPI frames, after an initial motion realignment step. Using this volumetric EPI as a target, the head motion parameters are estimated by realigning each EPI frame to the target using a rigid registration. To correct for EPI susceptibility distortions, the volumetric EPI is first subjected to an inhomogeneity correction step and then registered non-linearly to the anatomical scan from the same MRI session, which allows the calculation of the required geometrical transforms for recovering brain anatomy[42]. This data-driven approach to distortion correction was opted for as a default technique to maximise accessibility, like fMRIPrep, since it does not require additional acquisitions which may not be available (e.g., B0 field map or reversed phase encode image). Finally, after calculating the transformations required to correct for head motion and susceptibility distortions, both transforms are concatenated into a single resampling operation (avoiding multiple resampling) which is applied at each EPI frame, generating the preprocessed EPI timeseries in native space[7]. Preprocessed timeseries in common space are also generated by further concatenating the transforms allowing resampling to the reference atlas.

The preprocessing workflow of RABIES is illustrated in Fig. 1a, and each associated module is described in detail in the RABIES documentation (https://rabies.readthedocs.io/en/stable/). When structural images are not provided by the dataset, the volumetric EPI replaces the structural image during the generation of the unbiased template.

### Datasets for preprocessing benchmarking

The robustness of RABIES preprocessing was evaluated by sampling a range of datasets from different sites to attempt to capture representative data samples across the rodent fMRI community. We accessed 20 mouse fMRI datasets (totalising 318 C57Bl/6 J mice including both males and females), as well as 3 independent rat fMRI datasets (totalising 166 rats including Wistar (both male and females), Long-Evans (male only), and Sprague-Dawley (male and females) strains). No sex-specific analyses were conducted since the information in this regard was incomplete in the original publication of the datasets. All datasets are openly accessible, and acquisition details for each dataset can be found in their respective original publication (see Supplementary Table 2). All data was acquired with approval from local ethical authorities.

For mouse preprocessing using structural scans, we used the DSURQE ex-vivo MRI atlas[43–46], and the Fischer 344 in vivo atlas[47] was used for rat preprocessing. 17 mouse datasets did not include structural scans (see Supplementary Table 2), allowing for testing the workflow without structural scans, and for which we generated a mouse EPI atlas for the preprocessing of mouse datasets lacking structural images (see Methods section 3 below).

### Generation of a EPI atlas

We selected 5 mouse datasets (i.e., 117_Cryo_mediso_v, 7_Cryo_med_f1, 7_Cryo_med_f2, 7_Cryo_aw_f, 7_RT_med_f in Supplementary Table 2) from the multi-site study by Grandjean and colleagues[9]. We selected those datasets for their complete brain coverage, minimal susceptibility distortions and variable anatomical contrast. Using the RABIES workflow, each EPI scan was first corrected for inhomogeneity and an unbiased template was generated combining all 5 datasets. The unbiased template was then registered to the DSURQE mouse atlas using a non-linear registration[40] to propagate brain masks and anatomical labels onto the newly generated EPI template. This EPI atlas is openly available online (https://doi.org/10.5281/zenodo.5118030), and used by default by RABIES as reference atlas when executing the EPI-only preprocessing workflow.

### Preprocessing quality control

During preprocessing, a set of images is generated to conduct visual quality control of each failure-prone preprocessing step (Supplementary Fig. 1). To visualise the intensity inhomogeneity correction of either the structural or the EPI image, the input MRI image is shown before and after the correction together with the overlap of the brain mask used for the final correction, thus allowing to visualise the quality of inhomogeneity correction. Failures can be noticed either if important inhomogeneities remain or if brain masking fails. Then for the remaining registration steps, the report shows the overlap between the moving image and the target image. These registration steps include alignment to the generated unbiased template, the alignment between the unbiased template and the reference common space atlas, and the registration between anatomical and functional images for susceptibility distortion correction.

The entire quality control report from each dataset was visually inspected to record the number of failures at each registration step. Registration failures were considered when there was a clear misalignment of at least one brain structure. Minimal misalignments that subsisted because of EPI distortions and poor anatomical contrast were tolerated, given that few EPIs have sufficient contrast to achieve specific anatomical alignment and complete correction for distortions through non-linear registration. In Supplementary Table 2, additional notes on preprocessing quality are provided to detail dataset-specific limitations (e.g., partial misalignment to the reference atlas in highly distorted regions). All reports inspected are provided in this paper in the supplementary files.

### Confound correction workflow

The workflow for confound correction is structured following best practices found in human literature, largely based on recommendations from Power and colleagues[27]. Here, we detail the execution of the confound correction workflow, which requires specific considerations regarding the order and combination of multiple correction steps. These considerations aim to avoid inefficient corrections, or the reintroduction of confounds into the data at later steps[28]. Several

options are provided, and, if selected, are applied in the following sequence:

1.  Frame censoring: Frame censoring temporal masks are derived from framewise displacement and/or DVARS thresholds, and applied first on both BOLD timeseries before any other correction step to exclude signal spikes which may bias downstream corrections, in particular, detrending, frequency filtering, and confound regression[27].
    a. Censoring with framewise displacement: Apply frame censoring based on a framewise displacement threshold. The frames that exceed the given threshold, together with 1 back and 2 forward frames, will be masked out[17].
    b. Censoring with DVARS: The DVARS time course is z-scored (i.e., subtract the temporal mean and divide by the temporal standard deviation), and frames with values above a threshold of 2.5 (i.e., above 2.5 standard deviations from the mean) are removed. z-scoring and outlier detection is repeated within the remaining frames, iteratively, until no more outliers are detected, to obtain a final set of frames post-censoring. This second censoring strategy was implemented to address residual spiking artefacts which were not well detected with a one-shot framewise displacement censoring approach (Supplementary Fig. 17).
2.  Detrending: Linear (or quadratic) trends are removed from the timeseries. Detrended timeseries are obtained by performing ordinary-least square (OLS) linear regression, where regressors consist of time together with an intercept (time-squared is also included as a regressor if removing quadratic trends).
3.  ICA-AROMA: Motion sources can be cleaned using a rodent-adapted ICA-AROMA classifier[48] (Methods section 6). ICA-AROMA is applied prior to frequency filtering to remove further motion effects that can result in ringing after filtering[26,48].
4.  Frequency filtering: For highpass and/or lowpass filtering of timeseries, censored timepoints are first simulated while presenting the frequency profile of the timeseries to allow for the application of a Butterworth filter.
    a. Simulating censored time points: frequency filtering requires particular considerations when applied after frame censoring since conventional filters cannot handle missing data (censoring results in missing time points). To address this issue, we implemented a method described in Power et al.[27], allowing the simulation of data points while preserving the frequency composition of the data. This method relies on an adaptation of the Lomb-Scargle periodogram, which allows estimating the frequency composition of the timeseries despite missing data points, and from that estimation, missing time points can be simulated while preserving the frequency profile[49].
    b. Butterworth filter: Following the simulation, frequency filtering is applied using a 3rd-order Butterworth filter, and timepoints can be removed at each edge of the timeseries to account for edge artefacts following filtering[27]. Edge artefacts were visualised in simulated data (Supplementary Fig. 18), and from those observations, we conclude that removing 30 s at each edge and using a 3rd order filter is optimal for a highpass filter at 0.01 Hz. After frequency filtering, the temporal mask from censoring is re-applied to remove simulated time points.
5.  Confound regression: For each voxel timeseries, a selected set of nuisance regressors (Supplementary Table 5) are modelled using OLS linear regression and their modelled contribution to the signal is removed.

$$\boldsymbol{\beta} = \mathrm{OLS}(\mathbf{X}, \mathbf{Y})$$
$$\mathbf{Y_{CR}} = \mathbf{X}\boldsymbol{\beta} \qquad (1)$$
$$\hat{\mathbf{Y}} = \mathbf{Y} - \mathbf{Y_{CR}}$$

Regressed timeseries $\hat{\mathbf{Y}}$ are obtained with Eq. (1) where $\mathbf{Y}$ is the timeseries, $\mathbf{X}$ is the set of nuisance timecourses and $\mathbf{Y_{CR}}$ is the confound timeseries predicted from the model at each voxel ($\mathbf{Y_{CR}}$ is a time by voxel 2D matrix). Prior to the regression, the nuisance timecourses were subjected to the same frame censoring, detrending and frequency filtering, which were applied to the BOLD timeseries to avoid the re-introduction of previously corrected confounds[27,28].

6.  Intensity scaling: Signal amplitude must be scaled to allow for comparison of network amplitude estimates between scans[32,50]. The following options are available within RABIES:
    a. Grand mean: Timeseries are divided by the mean intensity across the brain, and then multiplied by 100 to obtain percent BOLD deviations from the mean. The mean intensity of each voxel is derived from the linear coefficient ($\beta$) from the intercept computed during detrending.
    b. Voxelwise mean: Same as grand mean, but each voxel is independently scaled by its own mean signal.
    c. Global standard deviation: Timeseries are divided by the total standard deviation across all voxel timeseries.
    d. Voxelwise z-scoring: Each voxel is divided by its standard deviation (i.e., z-scoring).
7.  Smoothing: Timeseries are spatially smoothed using a Gaussian smoothing filter.

Importantly, each confound correction step (with the exception of linear detrending) is optional when using RABIES to allow for adapting the correction strategy to specific dataset needs.

### Rodent-adapted ICA-AROMA
The original code for the algorithm (https://github.com/maartenmennes/ICA-AROMA) was adapted to function without the hard-coded human priors for anatomical masking and parameter thresholds for component classification. Following an initial independent component analysis (ICA) decomposition of the data using FSL's MELODIC algorithm[51], four features are extracted from each ICA component spatial map for classification: (1) the temporal correlation between the component and motion parameters, (2) the fraction of the frequency spectrum within the high-frequency range, and the fraction of the component within (3) the cerebrospinal fluid (CSF) mask and (4) the brain edge mask. The component is classified as motion if the CSF or high-frequency content fractions are above a given threshold, or if classified by a pre-trained linear classifier combining the brain edge fraction and motion correlation. To adapt the original algorithm with RABIES, the CSF mask is inherited from the rodent reference atlas and the edge mask is automatically generated from the brain mask, the threshold for high frequency content was increased as rodent can express higher BOLD frequencies, particularly under medetomidine[52], and the linear classifier was retrained. To select the new parameters, we manually classified motion and network components derived from a set of scans from the REST-AWK group anaesthetised under a medetomidine-isoflurane mixture, and selected parameters to successfully classify clear motion components while avoiding false classification of brain networks.

### Standardisation of variance voxelwise
Observing the distribution of signal variability across the brain (i.e., BOLD variability) allows one to identify confound signatures if present, whereas relatively uncorrupted scans present a predominantly homogeneous distribution of signal variability. To attempt mitigating the impact of confounds observed from the BOLD variability map (Fig. 2), we implemented a correction strategy to enforce a homogeneous variance distribution across voxels. To do so, timeseries are first scaled voxelwise by their standard deviation (yielding

standardised variance), and then timeseries are re-scaled to preserve the original total standard deviation of the entire 4D timeseries (i.e., the global standard deviation does not change). This is to avoid changing the overall scale of image intensity, which is handled separately during confound correction (see Methods section 5). Importantly, this is different from z-scoring, where timeseries are standardised to unit variance. Enforcing standardised variance distribution effectively downscales the contribution of voxels with higher variance, which typically corresponds to a confound signature.

### Connectivity analysis implemented within RABIES

Here we detail how each connectivity analysis available within RABIES can be conducted following the completion of the confound correction workflow.

Seed-based connectivity[29]: An anatomical seed of interest is selected, and the mean time course within the seed is correlated (Pearson's r) with each voxel timeseries to measure the brain connectivity map for that seed. In RABIES, this is repeated for each scan independently, to obtain a subject-level connectivity map that can then be used for hypothesis testing.

Whole-brain connectivity matrix[30]: the whole brain parcellation during RABIES preprocessing is used to extract a mean time course at each parcel. Then, all parcel timecourses are cross-correlated (Pearson's r) to obtain a whole-brain matrix describing the connectivity strength between every region pair. As with SBC, in RABIES, a connectivity matrix is computed for each scan.

Group independent component analysis (ICA)[51]: the timeseries from all scans are concatenated in common space, and ICA decomposition is conducted on those concatenated timeseries to identify spatial sources of covariance (components) prevalent in the entire group. Among those components, resting-state networks as well as sources of confounds can be identified (example in Supplementary Fig. 19). RABIES uses FSL's MELODIC ICA algorithm[51].

Dual regression[31,32]: Dual regression is a technique building on the group ICA algorithm which allows the modelling of previously detected group-level components back onto single scans. The dual regression algorithm consists of two consecutive linear regression steps. First, components from group ICA are regressed against an individual scan timeseries to obtain an associated time course for each component. Then, a second regression using these component timecourses is regressed again against the scan's timeseries, allowing to finally derive scan-specific spatial maps for each corresponding ICA component. Those linear coefficients can allow for describing individual-specific features of resting-state networks, such as amplitude or anatomical shape. To include network amplitude, the timeseries of the first regression are variance-normalised prior to the second regression, to weight the variance explained by each component into the spatial maps[32]. This method is applied to all dual regression analyses in this paper.

### Datasets included for data quality characterisation and guidelines

To explore signatures of confounds in mouse fMRI, we first relied on a dataset with a group of anaesthetised mice under a medetomidine-isoflurane mixture equipped with mechanical ventilation (i.e., controlling for motion) ($N = 10$ females) and a group of physically-restrained, freely-breathing, awake mice ($N = 9$ females)[53]. The group of awake mice was conditioned to the restraint system for 1 week prior to scanning. fMRI (gradient echo EPI, $0.18 \times 0.15$ mm and $90 \times 60$ matrix in-plane, 28 0.4 mm slices, TR = 1.2 s, 180 volumes) were acquired on an 11.7 T Bruker animal scanner. For each subject, data was acquired over 2 sessions where 3 consecutive fMRI scans were acquired per session, giving a total of 60 and 53 functional scans for the anaesthetised and awake group, respectively (one awake scan was lost during registration quality control). Furthermore, to benchmark

the quality control metrics and guidelines across a representative range of data quality found with mouse fMRI, we further leveraged a multicenter dataset previously harmonised by Grandjean and colleagues (17 acquisition sites, $N = 15$ scans per site, including both males and females)[9]. All datasets were preprocessed using only EPI scans, as the multicenter study did not provide structural scans.

### Group-ICA priors for dual regression

The group-ICA components used as priors for dual regression (available online https://zenodo.org/record/5118030/files/melodic_IC.nii.gz) were obtained using FSL's MELODIC algorithm[51]. MELODIC was run with 30 components on the combined data from the anaesthetised and awake mouse groups of the REST-AWK dataset, where confound correction consisted of highpass filtering at 0.01 Hz, FD and DVARS censoring, and confound regression was applied with the 6 rigid motion parameters together with white matter (WM)/CSF mask signals. The group-ICA maps were visually inspected, and inspired by suggestions from Zerbi and colleagues[19], each component was classified into a rodent brain network, confound source or other (Supplementary Fig. 19). More specifically, confound components display signatures of motion (i.e., edge effects), bilateral asymmetry, high loadings into WM and CSF tissues, widespread effects across the brain or affect only single slices. We opted for a conservative categorisation of components into confounds by selecting those which primarily displayed anatomically recognisable confound features without network-like features. For network analyses in this study, we selected two components corresponding to canonical mouse networks: (1) the somatomotor network, primarily expressed along the somatosensory and motor cortices[9,19], and (2) the default mode network, primarily expressed along the cingulate and retrosplenial cortices[9,19].

### Confound correction strategies investigated

To assess how confound correction can improve the quality of network analysis, the REST-AWK dataset and the 17 datasets from the multi-site study[9] were processed over a range of confound correction strategies. The datasets were initially corrected by applying a framewise displacement threshold of 0.05 mm and regressing the 6 motion parameters to first establish a baseline with minimal correction.

We then evaluated the impact of the following additional corrections: voxelwise variance standardisation (Methods section 7), DVARS censoring, 24 motion parameters regression[25], WM and CSF signal regression, aCompCor regression of the first 5 principal components, global signal regression, highpass filtering and ICA-AROMA. Lowpass filtering was not considered in this study, as lowpass inflates temporal autocorrelations[54] and impacts the measures of confound correlations at the scan level. Lowpass can be used in conjunction with the proposed framework in future studies but would require re-evaluating an appropriate threshold separately.

In all conditions, we applied linear detrending, grand mean scaling and spatial smoothing using a 0.3 mm Full-Width at Half Maximum Gaussian filter. When highpass filtering was applied, 30 s were removed at each end of the timeseries to avoid edge artefacts. After censoring time points, scans which had less than 2/3 of the original number of frames were removed from further analysis.

### Connectivity analyses and definition of canonical network maps

In this study, we consider the analysis of two different mouse canonical networks, namely the somatomotor and default mode networks[9,19,55], and each network was measured with either dual regression or seed-based connectivity. For seed-based connectivity analysis, the somatomotor and default mode networks were mapped using a seed in the right primary somatosensory cortex and anterior cingulate cortex, respectively. For dual regression, both networks were found among the group ICA components used as priors (Methods section 10).

To conduct the proposed analysis quality control framework, we needed to define a reference brain map for each canonical network and

each analysis technique. For dual regression, a representative map of each network was already defined from the group ICA priors. However, for seed-based connectivity analysis (Supplementary Figs. 15, 16), we needed to compute a representative average brain map for each network from the set of scans across datasets. To do so, we selected a subset of the datasets with various acquisition parameters (for the somatomotor network: 7_Cryo_aw_f, 7_Cryo_mediso_v, 7_RT_halo_v, 94_RT_iso_v, REST-AWK mediso; for the default mode network: 7_Cryo_mediso_v, 7_RT_halo_v, 94_RT_iso_v, 117_Cryo_mediso_v, REST-AWK mediso) and performed well on quality metrics with dual regression analysis, when using the initial confound correction strategy (framewise displacement censoring + 6 motion parameters regression). The consensus network map was obtained by taking the median connectivity across scans. The selection of datasets was repeated independently for each network to select datasets succeeding quality control for the relevant network. The resulting consensus network maps were highly similar to the corresponding ICA components.

### Dataset distribution report

To complement the group statistical report during analysis quality control (Fig. 4), as well as the selection of scan-level thresholds for network specificity and confound correlation (Fig. 3), an additional figure is generated to display the sample distribution across quality metrics (Supplementary Fig. 7). This report is automatically generated along the spatiotemporal diagnosis and group statistical reports and allows visualising the relationship of both network specificity and amplitude with each measure of confound, i.e., the within scan temporal correlation with confounds, mean framewise displacement, the variance explained from confound regression and temporal degrees of freedom. For dual regression only, the overall network amplitude is summarised for each scan by taking the L2-norm across the spatial map of network amplitude.

This report can be used to assess the proportion of scans which pass quality control thresholds, as well as evaluate the impact of modifying confound correction strategy on network specificity and temporal correlation with confounds. In addition, visualising the relationship between network amplitude and the three measures of confounds from the group statistical report allows for inspecting the correlational structure between connectivity and confounds. In particular, it can be useful for detecting outliers, which are automatically identified using a threshold of 3.5 on the modified Z-score[56] for each measure independently (network amplitude/shape, and each confound measure). Z-scores are computed on the subset of scans which passed scan-level quality control thresholds and after removing outliers in network amplitude (Methods section 14).

### Outlier removal based on network amplitude

Scans presenting outlier values in overall network amplitude from the distribution plot were automatically removed, as this likely indicates spurious connectivity (e.g., Supplementary Fig. 8). After removing scans which did not pass quality control, outliers were automatically detected and removed using the modified z-score, with a threshold of 3.5[56]. If outliers were detected and removed, z-scores were re-computed on the remaining scans, and new outliers were removed. This process was repeated iteratively until no more outliers were detected to obtain a group distribution without outliers.

### Confound correction optimisation protocol

We designed a stepwise protocol for optimising confound correction on a per-dataset basis, which refers to a table relating data quality features to the corresponding correction strategy (Supplementary Fig. 14a). For all analyses, we selected the following scan-level quality control thresholds: minimum of 0.4 Dice overlap for network specificity and maximum of 0.25 correlation for confound effects (Fig. 3). The procedure applied for each dataset was as follow:

1. The dataset is initially corrected only using framewise displacement censoring and regression of 6 motion parameters, and the initial data quality reports are generated.
2. Evaluation of data quality reports:
   a. Using the spatiotemporal diagnosis, the 4 main data quality markers described in Fig. 2 are inspected across scans to identify potential issues.
   b. The dataset distribution report is inspected to assess how many scans pass the quality control thresholds and identify outliers. If fewer than 8 scans passed, the group statistical report was not considered.
   c. Inspect the group statistical report to notice absent or spurious signatures in the network variability map and evaluate the extent of confound correlation (the average correlation within the network is saved in an attached CSV file).
3. An appropriate CR strategy is selected based on the observations from the data quality reports, following the order of priority and instructions in Supplementary Fig. 14a.
4. One additional correction is applied at a time to evaluate the impact of individual corrections. The quality outcomes are re-evaluated as in (2), and the correction is kept only if improving quality outcomes.
5. Repeat (3), (4) until no quality issues are left, or CR options are exhausted.

When evaluating the quality reports, the somatomotor - dual regression analysis was prioritised, but other analyses were also considered for improvements. The optimised set of corrections defined for each dataset is documented in Supplementary Fig. 14b.

### Reporting summary

Further information on research design is available in the Nature Portfolio Reporting Summary linked to this article.

## Data availability

All data used for this work is publicly available through the OpenNeuro[57] data-sharing platform (with the exception of the NITRC rat, provided through the NITRC platform). The online link to each dataset is found in sup. table 2. Additionally, supplementary files which accompany the results presented here are accessible at this online repository https://doi.org/10.17605/OSF.IO/GT7EX. Source data is also provided for Figs. 3, 5, and Supplementary Figs. 12b, 13, 15, 16. Source data are provided with this paper.

## Code availability

All code related to the RABIES software is openly accessible on GitHub (https://github.com/CoBrALab/RABIES). Installation and version control for all dependencies are managed through Docker containers, all readily available on the GitHub container registry for download (https://github.com/CoBrALab/RABIES/pkgs/container/rabies). In this work, preprocessing was conducted with version 0.3.3 for datasets with anatomical scans and 0.4.7 for preprocessing using only EPI (for results in Supplementary Table 2). For all results involving connectivity analyses, the pipeline steps for confound correction and analysis were conducted with version 0.5.0, while using the pre-generated preprocessing outputs from version 0.4.7. Of note, the same core registration algorithms are preserved across all software versions, and thus the same preprocessing outcome is expected. The software is predominantly written using Python and uses several packages, including NumPy v1.20.1, SciPy v1.6.2, NiPype v1.6.0, SimpleITK v2.0.2, NiBabel v3.2.1, Nilearn v0.7.1, pandas v1.2.4 and PyBIDS v0.13, as well as a modified version of ICA-AROMA (Methods section 6). Additional software dependencies used within the RABIES software include ANTs[58] v2.5.0, minc-toolkit v1.9.18, FSL[59] (using MELODIC version 3.14), and AFNI[60] 23.1.09. Instructions and associated scripts for reproducing

the RABIES outputs used in this paper can be found in a separate online repository (https://github.com/Gab-D-G/RABIES_paper_repro)[61]. The repository also regroups custom code for generating figures in this manuscript, which was done with a Python environment regrouping the Python packages listed above together with the RABIES python package.

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

## Acknowledgements

We thank the fMRIPrep team for their leadership in providing standards and guidelines in the development of open-source neuroimaging software. We also thank the ICA-AROMA developer for making their code open-access, which allowed us to integrate the algorithm with minor adaptations. We are grateful to all members of the mouse multicenter initiative from Grandjean et al.[9] as well as other members of the rodent fMRI community who publicly shared data. Without these recent initiatives, this work would not have been possible. We also thank Dr. Alessandro Gozzi and his laboratory Mila Urosevic for providing critical feedback regarding the manuscript, Daniel Gallino for suggesting the software acronym, and the RABIES users who provided feedback to correct issues and enhance the distribution of the software. Finally, we acknowledge our funding sources. The Fonds de recherche du Québec – Nature et technologies (FRQNT) scholarship (file #288131) for providing salary support to G.D.-G., the Dutch Research Council grant OCENW.KLEIN.334 supports the work of J.G., and M.M.C. receives salary support from the les Fonds de recherche du Québec – Santé (FRQS). Furthermore, M.M.C. research is supported by McGill University's Healthy Brains for Healthy Lives (a Canada First Research Excellence Fund programme), Canadian Institutes of Health Research, the National Sciences and Engineering Research Council of Canada, and a donation from the Toronto Dominion Bank.

## Author contributions

G. D.-G. led the project, analyses, and manuscript writing. G. D.-G wrote the code behind the RABIES software and all its associated documentation. G.A.D. contributed key preprocessing functions, in particular registration scripts, and supervised software design and distribution. G.A.D. is also the main developer of the Docker container. J.G. participated extensively in testing the software, provided recommendations and feedback regarding user needs, and contributed to software dissemination. J.G. also provided recommendations for selecting datasets in this study. M.M.C. supervised the research project and manuscript and supervised G. D.-G. G.A.D., J.G., and M.M.C. all provided feedback on analyses and contributed to editing the manuscript.

## Competing interests

The authors declare no competing interests.
