## [Peer Review File · Nature Communications]

nature portfolio

Peer Review File

A standardized image processing and data quality platform for rodent fMRI.Reviewer #1 (Remarks to the Author):

This manuscript presents a standardized image processing and data quality platform for rodent fMRI.

The development of such a platform is highly topical and much needed in rodent fMRI community. It comes as an important building block following recent guidelines in rodent fMRI acquisition practices (Grandjean et al, Nat Neurosci 2023).

The manuscript is very well written and clear.

There are a few points I would encourage the authors to consider:

- The RABIES software has been available and used for a few years now, including in the (Grandjean et al, Nat Neurosci 2023) paper. Perhaps it would be helpful to clarify the timing and scope of the current manuscript in the context of the software versioning history.
- Is the customizable nature of the confound correction strategies expected to improve comparability of datasets across sites or rather challenge it? What is the fine line between optimizing rather than tweaking to obtain an a priori expected outcome? What would be the impact of applying a comprehensive list of confound corrections without customization? In principle, if a confound is not heavily present, correcting for it should not affect the data too much, or would it?
- How were the cut-offs in Fig. 3 determined? (0.4 Dice overlap and 0.25 temporal correlation with confounds) Were they dependent on the datasets included in the study? If completely unseen datasets were now run through the pipeline (i.e. some sort of test data) would the thresholds be reliably applicable to those new datasets?
- What about studies of rodent models of disease? I.e. if some RSNs can be expected to be affected / attenuated etc?
- Should RABIES accommodate reversed phase encode images for EPI distortion correction? How is the issue of irrecoverable signal dropouts managed?

Reviewer #2 (Remarks to the Author):

The paper authored by Desrosiers-Gregoire and colleagues endeavors to establish standardized and meticulously refined data preprocessing pipelines for rodent functional MRI, accompanied by a rigorous quality control procedure and correction for confounding artifacts.

The open-access algorithms presented by Rodent Automated Bold Improvement of EPI Sequences (RABIES) aspire to enhance reproducibility and transparency in the field. Addressing the vital aspects of quality control and advocating best practices for data preprocessing and analysis holds significant importance at the current stage for the community of preclinical imagers.

A key innovation in this work is the testing of pipelines using a large dataset recently acquired from two multi-center studies in both mice and rats. This study stands out as the first of its kind to apply such pipelines in extensive cohorts, validating the algorithms with data from multiple centers worldwide, encompassing different scan protocols and magnetic fields.

Overall, the paper is well-written, providing a detailed and comprehensive description. The step-by-step procedures are also thoroughly documented online.

My considerations and recommendations primarily focus not on the content, which I commend the authors for, but rather on the genuine novelty and enhancements introduced by this work, along with the authors' long-term strategy for sustaining this platform. Additionally, I have included a few minor suggestions.

1) Software Maintenance Considerations. In line with numerous articles introducing novel pipelines, it becomes pivotal to delve into the community's prospective acceptance of the presented work. While the arguments pertaining to analyses and their presentation are compelling, a significant lacuna arises concerning the long-term maintenance strategy for the introduced software. The pivotal question emerges: How does the author intend to sustain and update the software over time? Are we susceptible to being entrenched with an unchanging pipeline and software, bereft of future updates? Alternatively, is there a concrete plan to evolve RABIES into a

stable entity that would not depend on singular individuals or lab? In my experience, I have witnessed the emergence of promising software solutions that, despite initial adoption, faltered due to a lack of sustained maintenance. Although this information may possess marginal scientific relevance, its inclusion is indispensable for the project's enduring success.

2) The authors assert in the results section: "We achieved a 100% (255/255) success rate for the inhomogeneity correction of EPIs, 99.6% (254/255) success during cross-subject alignment to the unbiased template." This declaration may pose challenges for an MRI expert to fully endorse. Inhomogeneity correction is an inherent characteristic of fast echo planar imaging, particularly for gradient-echoes, influenced by the rapid dephasing of spins due to various factors (e.g., air-tissue interfaces near the ear channel). True correction typically involves employing experimental techniques such as acquiring the same image with opposite gradient phase encoding. In post-processing steps, as in this case, asserting a 100% success rate for inhomogeneity correction in EPIs might be better framed as achieving acceptable registration/normalization (within defined thresholds) and correction for the B1 bias field. I would encourage the authors to reconsider the wording in this section.

3) The authors humbly acknowledge the recent introduction of other rodent-adapted software pipelines for mice and rats (Celestine et al., 2020; Diao et al., 2021; Ioanas et al., 2021; Lee et al., 2021). However, they assert that no existing software has undergone comprehensive validation across acquisition sites and species concerning preprocessing quality. While recognizing the presence of other pipelines, the claim that these are only insufficiently tested (as their main weakness) suggest little novelty in the current work. To strengthen this argument, the authors should provide more compelling evidence and quantifiable metrics, such as comparing RABIES results against each of these other strategies, to demonstrate the superiority and ease of adaptation of their protocol for the majority of labs.

4) The authors could include and discuss another recent reference that aimed to provide a standardised pipeline for an optimized registration workflow in both humans and animals (including rodents (PMID: 37090033)).

5) The authors employ a quality control method similar to the one introduced in a recent multi-site rat paper for resting-state fMRI. However, the novelty of this approach in the current study remains unclear. It would be beneficial for the authors to provide a more extensive discussion or comparison with existing methodologies to elucidate the distinctive contributions and advancements their chosen quality control approach brings to the field.

We thank the reviewers for their comments and feedback. Excellent points were raised, and we believe the clarity and impact of the manuscript will be improved by addressing the reviewer feedback. We hope our edits adequately address those points, and that our revised manuscript will meet standards for *Nature Communication*. In the following text, we provide our responses in *italic font*, and citations from the revised manuscript are “*within quotation marks*”. Our revisions are highlighted in “*purple font*”. The comment from each reviewer is shown in plain text.

Reviewer #1 (Remarks to the Author):

This manuscript presents a standardized image processing and data quality platform for rodent fMRI. The development of such a platform is highly topical and much needed in rodent fMRI community. It comes as an important building block following recent guidelines in rodent fMRI acquisition practices (Grandjean et al, Nat Neurosci 2023). The manuscript is very well written and clear.

There are a few points I would encourage the authors to consider:

- The RABIES software has been available and used for a few years now, including in the (Grandjean et al, Nat Neurosci 2023) paper. Perhaps it would be helpful to clarify the timing and scope of the current manuscript in the context of the software versioning history.

We appreciate the reviewer noticing that the RABIES package was first used in the Grandjean et al. Nat Neurosci paper. However, the implementation of the package, at the time, was not thoroughly described and was mainly used for the purpose of its robust preprocessing capabilities across datasets. Here, the software is formally introduced and described across all its features and intended usage. To this end, we fully describe the technical implementation, further the validation criteria across both mice and rat datasets (the original paper was done in the context of rat fMRI studies), and introduce a full suite of quality control reporting tools and quality assessment criteria that were not at all mentioned in the original paper. This is supported by the software version for the pipeline originally used in the Grandjean et al paper (v0.3.5) and the one that is presented here (v0.4.8).

We have added a new comment in this respect at the end of the introduction:

“Inspired from previous initiatives fostering reproducibility and standardization in humans (Craddock et al., 2013; Esteban et al., 2019; Taylor et

al., 2023), these innovations are integrated through the distribution of an open-source software platform, Rodent Automated Bold Improvement of EPI Sequences (RABIES; publicly available to the community <https://github.com/CoBrALab/RABIES> and thoroughly documented online <https://rabies.readthedocs.io/en/stable/>) that provides an integrative solution for preprocessing, confound correction, analysis, together with data diagnostic tools encouraging best practices for quality control. While the RABIES pipeline was used elsewhere for preprocessing in rats (Grandjean et al., 2023), in the current manuscript we formally describe the software's technical implementation, its validation across rodent species, and develop a novel suite of data quality reports and guidelines tailored for tackling comparability between rodent datasets."

It is also important to highlight a departure from the quality control framework used in the previous studies from Grandjean and colleagues. In Grandjean et. al. data quality was conducted with a seed to seed connectivity framework between primary sensory cortex (S1) and anterior cingulate cortex (ACC) as biological priors, whereas our proposed framework using Dice overlap is data-driven for generalizability. This particular point is further discussed below in response to referee #2's question on this topic.

The following comment from the reviewer contained several sub-questions, and for simplicity and clarity, we respond to each sub-question as follows:

1. In principle, if a confound is not heavily present, correcting for it should not affect the data too much, or would it?

There can indeed be negative impacts to applying a correction when confounds are not present. To name a few instances: 1) the hotly debated global signal regression can remove signal of interest or introduce artifactual negative correlations (see Murphy & Fox, 2017 for a review of the debate), 2) corrections generally lead to loss in degrees of freedom, and 3) any regression (even using random regressors) can potentially remove network signal (see Bright & Murphy, 2015). We show in our own results that global signal regression may remove network signal in a high-quality scan (see **sup. figure 6B** and **sup. figure 9**). Confound correction is thus a matter of trade-off between sufficient correction, but not too much.

2. What would be the impact of applying a comprehensive list of confound corrections without customization?

*In this work, when we tried to identify best candidates for a universal correction (i.e. without customization) across datasets (**sup. figure 9**), we found that no individual technique can unequivocally provide only benefits across all data. Because of this initial observation, we expect the application of the same correction across datasets to be sub-optimal (i.e. would remove more signal than necessary in some datasets). In **sup. figure 9** we did not attempt complex combinations of techniques beyond ‘motion regression + frame censoring + one additional strategy’. It is due to this unpredictable impact of corrections that we instead recommend relying on QC outcomes to best judge which correction is suitable for a given dataset.*

3. - Is the customizable nature of the confound correction strategies expected to improve comparability of datasets across sites or rather challenge it?

Given our conclusion above that a fixed confound correction strategy across datasets challenges comparability, we indeed argue that optimizing the strategy to meet clear quality control standards is a better alternative. This is because QC measures directly relate to result interpretability. We designed the QC evaluation in this manuscript to relay minimal, comprehensive, and reasonable expectations for a connectivity study (network detectability and minimal confound correlation). Because we observed important divergences in these QC outcomes between datasets while applying a uniform correction, we argue that opting in this case for customized strategies improves the reconciliation of connectivity results.

4. What is the fine line between optimizing rather than tweaking to obtain an a priori expected outcome?

*It is true that there must be a fine line between improving confound correction and overfitting to these a priori expectations from QC (since of course QC metrics have their own limitations, see **sup. table 4**). For this reason, it remains important for researchers to be cognisant of the interpretation and limitations of QC metrics. When comparing multiple datasets, it would probably be preferable to select a consistent method across datasets if improvements are minor and inconsequential when tweaking the correction. This was not the case in the current manuscript.*

Based on the related questions listed above, we expanded the supplementary discussion relating to those decisions:

“Section 3: Variable impact of confound correction across datasets and the customization of correction strategy on a per dataset basis The impact of each confound correction strategies across datasets

To evaluate the ability of confound correction to improve analysis quality outcomes and investigate whether specific strategies can provide generalizable improvements across datasets, the main strategies available within RABIES were all tested systematically across the 19 datasets considered in part 2 of the manuscript (**sup. figure 9**). More specifically, datasets were initially processed with the regression of 6 motion parameters and censoring with framewise displacement, then one additional correction was applied. The impact of the correction was evaluated across quality metrics part of the quality control guidelines (**figure 3 & 4**). Outcomes were also evaluated when removing initial corrections (using only censoring, or no correction applied) to inspect potential impacts of the baseline correction and benchmark the ability of quality control metrics in detecting problematic quality outcomes.

Overall, there was important variability in the impact of additional corrections across datasets. Although certain corrections predominantly yielded improvements in quality metrics (i.e. standard, WM/CSF and aCompCor), no single correction led to uniform increase in the number of scans passing network specificity and confound correlation thresholds across all datasets. *This conclusion is based on the report shown in **sup. figure 9** for the somatomotor network analysis using dual regression, as well as similar reports generated for alternative connectivity analyses shared in the supplementary files.* The application of no correction (raw) or only framewise displacement censoring led to an almost uniform decrease in quality, thus justifying the recommendation of regressing 6 motion parameters and applying censoring as a baseline. ~~The impact of each correction as reported in **sup. figure 9** (together with similar reports for other connectivity analyses provided along supplementary files) supported defining recommendations for optimizing confound correction in **extended data figure 10** according to the quality metrics of network specificity and confound correlation.~~

*This observed variability is reminiscent of the fact that there is no consensus regarding an ideal correction strategy that should be applied across datasets and experiments (Power et al., 2020; Satterthwaite et al., 2019). This is why instead of providing an ideal strategy that should be blindly replicated, we opt for an approach grounded in an understanding of the data and associated research goals. In this case, we put forward guidelines describing how the data quality reports put forward in this manuscript can be leveraged to achieve standard connectivity analysis (**extended data figure 10**). We argue that such a framework is best for study comparison, over the replication of the exact same methodology, since the quality control measures relate more directly to analysis outcomes and goals.*

*It is important, however, to note that there can always be a risk of overfitting to meet a priori expectations (since of course quality control metrics have their own limitations, as listed in **sup. table 4**). For this reason, it remains important for researchers to be cognisant of the interpretation and limitations of those metrics. When comparing dataset, if there are minor and inconsequential dataset differences in the impact of confound correction on quality, it can become preferable to opt for methodological consistency. This was not the case in the current manuscript.”*

We also modified the following paragraph from the results section (paragraph 2 in ‘Applications of the analysis quality control framework for improving data quality’):

*“Ideally, the confound correction strategy is adapted to correct the specific issues present in a given dataset, given that excess correction can negatively impact data quality by removing network activity (Bright & Murphy, 2015; Power et al., 2020; Satterthwaite et al., 2019). This was tested by comparing the impact of various correction methods available within RABIES on the scan-level and group-level quality. We observed important trade-offs across different aspects of data quality (e.g. reduction of confound correlation at the cost of network detectability), and important variability between datasets regarding the effectiveness of a given correction strategy (**sup. figure 9**) ~~(**sup. material section 3**)~~. Thus, instead of applying the same strategy across datasets, we leveraged the spatiotemporal diagnosis and analysis quality control to define a principled approach to designing dataset-specific corrections which address relevant aspects of data quality (**methods described in methods section 15, and the rationale discussed in sup. material section 3**). With this approach, correction strategies are combined incrementally, evaluating scan quality features and analysis quality control at each iteration, with the goal of maximizing network detectability and minimizing spurious effects.”*

- How were the cut-offs in Fig. 3 determined? (0.4 Dice overlap and 0.25 temporal correlation with confounds) Were they dependent on the datasets included in the study? If completely unseen datasets were now run through the pipeline (i.e. some sort of test data) would the thresholds be reliably applicable to those new datasets?

The thresholds were selected upon visual inspection of the features described in figure 2, that is, the Dice threshold was selected to distinguish scans where network connectivity was specific instead of absent or spurious, and the confound correlation threshold was selected to delineate spurious scans. However, we do not expect these specific threshold values to generalize across all experiments, as various parameters

may influence the metrics (e.g. field of view, voxel resolution, SNR, or the application of lowpass filtering). Thus, we always would recommend instead to rely on visual inspection of the scan diagnosis to assess an appropriate threshold (as described in our online documentation

https://rabies.readthedocs.io/en/stable/analysis_QC.html#:~:text=If%20spurious%20or,%2D%2Dscan_QC_thresholds). In this manuscript, we selected a fixed threshold across datasets for consistency, and ensured that the thresholds were appropriate across the datasets considered through visual inspection.

To improve clarity, we edited the result section ‘A network-level quality control framework prior to statistical analysis.’ to include the following sentence:

“We propose that scans which do not meet thresholds on either measure be removed from downstream statistical analysis. *These thresholds should be selected by inspecting quality markers from **figure 2**, identifying scans with spurious or absent connectivity, and determining an appropriate threshold for each metric to delineate these scans. In addition to these two thresholds* Additionally, for dual regression analysis, scans which present outlier values in network amplitude are removed (**methods section 16**).”

We also added further details regarding these notions in the discussion (paragraph 3):

“Similar to the quality control of preprocessing, the proposed guidelines for the quality control of network analysis rely on the visual interpretation of reports comprising several metrics. This is necessary, as generic quality control metrics have their shortcomings (**sup. table 4**), *and their affiliated thresholds may not blindly generalize across all datasets. Instead, as highlighted in a recent editorial, the bedrock of quality control with fMRI still rests on qualitative assessment of the data, which can then support quantitative methods (Taylor et al., 2023). For this purpose, RABIES provides adequate automatically-generated visual reports to complement the interpretation of quality control metrics and the selection of appropriate thresholds.*”

- What about studies of rodent models of disease? I.e. if some RSNs can be expected to be affected / attenuated etc?

We believe this question is already addressed in the discussion (end of 3rd paragraph):

“Finally, although the quality control guidelines we present here can greatly support evaluating the main pitfalls in network analysis, these recommendations are not meant to be required irrespective of the scientific question and experimental design. The judgement of the experimenter remains paramount in determining which aspect applies to their study (for instance, network detectability may not always be expected, if studying the impact of anesthesia or inspecting a visual network in blind subjects). The guidelines are thus meant to offer a baseline which may apply to most standard resting-state network studies, as well as to instruct new researchers and RABIES users as to the main confounding factors which may impact analysis.”

The researchers’ judgement remains the main actor. If a network is expected to be absent, then it only makes sense to test for the presence of the network the control animal group. We would expect that for most studies of RSN in rodent models the network should be expected to be present, but this question is beyond the scope of the manuscript.

- Should RABIES accommodate reversed phase encode images for EPI distortion correction? How is the issue of irrecoverable signal dropouts managed?

As a default distortion correction technique, we selected a non-linear registration approach, as did the developers of fMRIPrep, since this approach does require additional acquisitions which may not be available (B0 field map or reversed phase encode image). This was thus a design decision on our part to maximize accessibility and application across existing datasets. However, we acknowledge that this approach may not achieve highest performance in all datasets (as reported in extended data figure 1), and may not be ideal for applications specifically interested in studying heavily distorted regions.

At the moment, we have not implemented alternative distortion correction techniques, but we would welcome user contributions in doing so. RABIES is an open project which will evolve according to the contributions and demands from the community. We have documented guidelines for supporting such contributions <https://rabies.readthedocs.io/en/stable/contributing.html>.

In response to a question from referee #2 regarding software maintenance, the discussion was modified to invite community involvement and mention contribution guidelines (see below). We also bring the following modifications to methods section 1, 3rd paragraph:

“The remaining preprocessing involves the EPI images. A volumetric EPI image is first derived using a trimmed mean across the EPI frames, after an initial

motion realignment step. Using this volumetric EPI as a target, the head motion parameters are estimated by realigning each EPI frame to the target using a rigid registration. To correct for EPI susceptibility distortions, the volumetric EPI is first subjected to an inhomogeneity correction step, and then registered non-linearly to the anatomical scan from the same MRI session, which allows to calculate the required geometrical transforms for recovering brain anatomy (Wang et al., 2017). This data-driven approach to distortion correction was opted for as a default technique to maximize accessibility, like fMRIPrep, since it does not require additional acquisitions which may not be available (e.g. B0 field map or reversed phase encode image). Finally, after calculating the transformations required to correct for head motion and susceptibility distortions, both transforms are concatenated into a single resampling operation (avoiding multiple resampling) which is applied at each EPI frame, generating the preprocessed EPI timeseries in native space (Esteban et al., 2019). Preprocessed timeseries in common space are also generated by further concatenating the transforms allowing resampling to the reference atlas.”

Reviewer #2 (Remarks to the Author):

The paper authored by Desrosiers-Gregoire and colleagues endeavors to establish standardized and meticulously refined data preprocessing pipelines for rodent functional MRI, accompanied by a rigorous quality control procedure and correction for confounding artifacts.

The open-access algorithms presented by Rodent Automated Bold Improvement of EPI Sequences (RABIES) aspire to enhance reproducibility and transparency in the field. Addressing the vital aspects of quality control and advocating best practices for data preprocessing and analysis holds significant importance at the current stage for the community of preclinical imagers.

A key innovation in this work is the testing of pipelines using a large dataset recently acquired from two multi-center studies in both mice and rats. This study stands out as the first of its kind to apply such pipelines in extensive cohorts, validating the algorithms with data from multiple centers worldwide, encompassing different scan protocols and magnetic fields.

Overall, the paper is well-written, providing a detailed and comprehensive description. The step-by-step procedures are also thoroughly documented online.

My considerations and recommendations primarily focus not on the content, which I commend the authors for, but rather on the genuine novelty and enhancements introduced by this work, along with the authors' long-term strategy for sustaining this platform. Additionally, I have included a few minor suggestions.

1) Software Maintenance Considerations. In line with numerous articles introducing novel pipelines, it becomes pivotal to delve into the community's prospective acceptance of the presented work. While the arguments pertaining to analyses and their presentation are compelling, a significant lacuna arises concerning the long-term maintenance strategy for the introduced software. The pivotal question emerges: How does the author intend to sustain and update the software over time? Are we susceptible to being entrenched with an unchanging pipeline and software, bereft of future updates? Alternatively, is there a concrete plan to evolve RABIES into a stable entity that would not depend on singular individuals or lab? In my experience, I have witnessed the emergence of promising software solutions that, despite initial adoption, faltered due to a lack of sustained maintenance. Although this information may possess marginal scientific relevance, its inclusion is indispensable for the project's enduring success.

We absolutely appreciate and share the reviewer's concern for sustainability. First and foremost, RABIES is completely open source and any user is welcome to add functionality, features, or to change any of the applications. We encourage user involvement and have written contribution guidelines for this purpose <https://rabies.readthedocs.io/en/stable/contributing.html>. Further still, the pipeline emerges from the CoBrA Lab, where the lead author (Desrosiers-Gregoire is a senior graduate student). Much of the work of this laboratory is focused on the development and dissemination of novel analytical pipelines and the maintenance of well-used ones. This includes a number of tools such as MAGEt Brain, minc-bpipe-library, optimized_antsMultivariateTemplateConstruction, qbatch, and others. Further, the laboratory employs a long term full-time research associate (G.A. Devenyi) who has played a central role in the maintenance, support, and distribution of these packages. Given the ongoing track-record of the group, we do not feel that there is a risk for users in the adoption of this pipeline.

We brought the following addition the discussion's last paragraph:

"We provide the community with a reliable tool for conducting image processing ~~and~~ together with recommendations for analysis quality control, thereby allowing harmonization of computational practices across laboratories and fostering reproducible research. This advancement is timely in light of current inconsistencies in methodological and reporting practices for rodent fMRI. Like previous neuroimaging softwares (Abraham et al., 2014; Ciric et al., 2018; Craddock et al., 2013; Esteban et al., 2017, 2019), RABIES is made open source to encourage community involvement in the development of additional features (see our online guidelines <https://rabies.readthedocs.io/en/stable/contributing.html>). This will support expanding beyond the default workflow presented here to address more specific needs (e.g. inclusion of alternative distortion correction methods or multi-echo preprocessing), and ultimately, involve the community in meeting evolving best practices over time. Previous initiatives, such as fMRIPrep, have already been transformative in those regards for human work. ..."

2) The authors assert in the results section: "We achieved a 100% (255/255) success rate for the inhomogeneity correction of EPIs, 99.6% (254/255) success during cross-subject alignment to the unbiased template." This declaration may pose challenges for an MRI expert to fully endorse. Inhomogeneity correction is an inherent characteristic of fast echo planar imaging, particularly for gradient-echoes, influenced by the rapid dephasing of spins due to various factors (e.g., air-tissue interfaces near the ear channel). True correction typically involves employing experimental techniques such

as acquiring the same image with opposite gradient phase encoding. In post-processing steps, as in this case, asserting a 100% success rate for inhomogeneity correction in EPIs might be better framed as achieving acceptable registration/normalization (within defined thresholds) and correction for the B1 bias field. I would encourage the authors to reconsider the wording in this section.

*To improve clarity, we re-wrote the paragraph from the ‘A generalizable registration workflow across rodent species and acquisition sites’ results section as shown below. Details regarding our criteria for what constitutes ‘success’ is found in the **methods section 4**, where it is mentioned that a partial misalignment was tolerated in the correction for susceptibility artefacts in some of the datasets, as these affected only a minimal set of brain regions.*

*“Each dataset was processed individually using the final workflow and performance was evaluated by inspecting the quality control report, where a ‘successful’ registration would provide sufficient image alignment across the brain for most standard analyses (see details in **methods section 4**). In the improved workflow, failures were marginal (extended data figure 1; sup. figure 1). For the 17 mouse datasets lacking structural images, we achieved a 100% (255/255) success rate for the **B1 intensity inhomogeneity correction step in EPIs** and 99.6% (254/255) success during cross-subject alignment to the unbiased template. For the 3 mouse datasets including supporting structural scans, we obtained 100% success rate for **intensity inhomogeneity correction in both structural (110/110) and EPI (185/185) scans inhomogeneity correction**. We achieved a success rate of 99.1% (109/110) for the alignment to the unbiased template and 100% (185/185) **for the susceptibility distortion correction of EPIs through non-linear alignment ~~cross-modal alignment of the EPI~~** to the structural scan. Finally, in the 3 rat datasets (1 dataset only included EPI scans), 98.9% (88/89) and 100% (230/230) of scans passed inhomogeneity correction for anatomical and EPI scans respectively, and we obtained 99.6% (230/231) success rate for cross-subject alignment and 98.9% (87/88) success rate during **susceptibility distortion correction ~~cross-modal alignment~~**. Detailed reports for each dataset can be found in extended data figure 1, and the visual quality control report from each preprocessing run is provided in the supplementary files. We consider that this high success rate can address the significant preprocessing needs within the rodent fMRI community, despite the current cross-site variability in acquisition equipment and parameters.”*

3) The authors humbly acknowledge the recent introduction of other rodent-adapted software pipelines for mice and rats (Celestine et al., 2020; Diao et al., 2021; Ioannas et al., 2021; Lee et al., 2021). However, they assert that no existing software has undergone comprehensive validation across acquisition sites and species concerning preprocessing quality. While recognizing the presence of other pipelines, the claim that these are only insufficiently tested (as their main weakness) suggest little novelty in the current work. To strengthen this argument, the authors should provide more compelling evidence and quantifiable metrics, such as comparing RABIES results against each of these other strategies, to demonstrate the superiority and ease of adaptation of their protocol for the majority of labs.

*To address this point, we undertook further investigation into how RABIES can be compared to other existing rodent fMRI pipelines. We considered every open source software project known to us at this moment (including QuNex, which is mentioned in a comment below from the referee), for a total of 6 different projects (1 is unpublished). Conducting a comprehensive, accurate and just comparison between pipelines is challenging. The initial key elements of an open source community tool concerns accessibility, documentation, maintenance and overall usability. We undertook a comparison of other pipelines along these considerations, and compiled our observations **in a table attached to this revision**. As can be appreciated from the table, alternative softwares are either A) unmaintained for a number of years, B) lack documentation, or C) are not distributed through a container or stable alternative for handling dependencies and reproducibility. The only exception is QuNex, but the rodent pipeline is only a subset of this initiative (QuNex is an agglomeration of resources for processing various MRI modalities across species), and as stated in its recent publication, this aspect of the software is still under development, lacking an official release, and was not tested on rodent data as part of the publication. Nevertheless, we identified three candidate software which were most accessible despite aforementioned limitations, SAMRI and Samba-MRI, and attempted their implementation. However, as documented in the table, we were unable to execute the pipelines without important barriers with regards to dependency management, despite the use of BIDS format and following available instructions (if any) for each respective software.*

This is in sharp contrast with RABIES' accessibility standards: 1) all dependencies are managed through Docker and Singularity containers, mitigating any user management, 2) includes BIDS compatibility, 3) user documentation is extensive, including a hands-on tutorial https://github.com/CoBrALab/RABIES_tutorial and a step-by-step pipeline workshop recorded from our institute https://www.youtube.com/watch?v=LZohKIUgycc&t=2766s&ab_channel=DouglasResearchCentre. The pipeline's accessibility can be further highlighted by our growing user

database, demonstrated by the ongoing activity on the Github discussion board and the number of container downloads now surpassing 3K (590 from github <https://github.com/CoBrALab/RABIES/pkgs/container/rabies> and 2.6K from the earlier Docker hub releases <https://hub.docker.com/r/gabdesgreg/rabies/tags>). As such, we conclude that the RABIES pipeline offers substantial advantage with regards to accessibility, an essential element for the widespread adaptation of standard practices for the rodent imaging community.

Given the aforementioned roadblocks with regards to software implementation, we consider a formal comparison performance to be beyond the scope of this manuscript. We would additionally emphasize that, outside of implementation challenges, providing an accurate and unbiased representation of another software, as an inexperienced user, is difficult. New users may not present the expertise to achieve top performance, and as competitors, our assessment may be biased. Instead, a co-author on this work, Dr. Grandjean, is currently leading an ideal alternative, where developers of rodent fMRI softwares were invited to use their respective pipeline for processing a common dataset ‘their way’, and share the resulting outputs for a systematic, unbiased comparison of software performance. This study was recently pre-registered (<https://osf.io/pmdge>).

To emphasize RABIES’ accessibility advantages, the following was added to the manuscript’s discussion (end of first paragraph):

“We introduce a robust image processing and analysis pipeline adapted for specific challenges to rodent fMRI. While previous preprocessing pipelines were introduced for rodent fMRI (Celestine et al., 2020; Diao et al., 2021; Ioannas et al., 2021; Lee et al., 2021), the strengths of the proposed approach include a thorough validation across acquisition sites and species, spanning broad differences in image quality, together with the integration of state-of-the-art tools for confound correction, analysis and data quality assessment. To date, RABIES is also the only rodent software combining containerized distribution, BIDS compatibility and extensive user documentation. These advances with regards to accessibility are essential for the widespread adaptation of standard practices across the rodent imaging community.”

4) The authors could include and discuss another recent reference that aimed to provide a standardised pipeline for an optimized registration workflow in both humans and animals (including rodents (PMID: 37090033)).

We added this reference to the list of rodent pipelines in the introduction paragraph 2:

“Although rodent-adapted software pipelines were recently introduced for either mice and rats (Celestine et al., 2020; Diao et al., 2021; Ioannas et al., 2021; Ji et al., 2023; Lee et al., 2021), no existing software has been thoroughly validated across acquisition sites and species in terms of preprocessing quality.”

And in the discussion, first paragraph:

“While previous preprocessing pipelines were introduced for rodent fMRI (Celestine et al., 2020; Diao et al., 2021; Ioannas et al., 2021; Ji et al., 2023; Lee et al., 2021), the strengths of the proposed approach include a thorough validation across acquisition sites and species, spanning broad differences in image quality, together with the integration of state-of-the-art tools for confound correction, analysis and data quality assessment.”

5) The authors employ a quality control method similar to the one introduced in a recent multi-site rat paper for resting-state fMRI. However, the novelty of this approach in the current study remains unclear. It would be beneficial for the authors to provide a more extensive discussion or comparison with existing methodologies to elucidate the distinctive contributions and advancements their chosen quality control approach brings to the field.

The metrics used in the current study versus the one used in previous work by Grandjean differ in their goals. In the former study by Grandjean, the suggested metric aimed to provide a simple, biologically grounded measure benchmarking connectivity relative to the main rodent networks. This choice was adequate in presenting an interpretable survey of data quality divergences across the whole set of datasets included in those studies.

However, limitations of this metric contrasted with the goals of the current study. In the current work, the primary goal is to provide tools which can generalize across a range of possible applications, i.e. for any possible region or network of interest, and to provide sufficient tools to adequately benchmark data quality for any single dataset. With this scope in mind, the metrics introduced in the current manuscript allow inspecting the specificity of network connectivity similarly to the previous metric by Grandjean, but without anatomical constraints (i.e. using Dice overlap to inspect network shape for any provided network). It is also complemented with additional measures ensuring a more complete benchmark of potential QC pitfalls (i.e. correlation with confounds, group statistical report...). This comes at the cost of simplicity, but provides an adequate toolkit for a range of potential applications.

The first paragraph of results part 2 was modified:

“... Here, we thus *expand on previous work in rodents (sup. material section 4) to define guidelines for network analysis quality control across two separate axes: spurious effects from confounds and network detectability. ...*”

And we wrote a added a supplementary discussion section on this topic:

“Section 4) Comparison of chosen quality control approach with previous methodology by Grandjean and colleagues

Grandjean and colleagues first introduced a quality control measure accounting for the presence and specificity of canonical rodent networks (Grandjean et al., 2020). The method consists in measuring two functional connections: 1) a positive correlation between seeds in the right and left somatosensory cortices (S1), and 2) a null or negative correlation between the S1 and anterior cingulate seeds. The former assesses the expected functional connectivity arising from the somatomotor network, and is supported by the presence of known anatomical connectivity. The latter is expected as the ACC belongs to the so-called task negative network, which tends to be anti-correlated with the somatomotor network, and there are no direct axonal connections between those regions. Using this framework, the authors conducted the first multi-site comparison of data quality in mice (Grandjean et al., 2020), and later in rats (Grandjean et al., 2023), where they delineated four types of scan quality outcomes: specific, absent, spurious or unspecific connectivity. This approach is well grounded in biological priors and common notions of canonical network connectivity, while remaining interpretable and easily implemented.

However, while this technique provides important advantages in contrasting connectivity features across datasets, the anatomical constraints of the technique contrasted with the goals of the current manuscript. The RABIES toolkit aims to provide generalizable methods across a range of possible applications by allowing 1) application of the methods to any network/region of interest and 2) capture spurious features which may arise in any brain region. For this purpose, the measure of network specificity through Dice overlap was introduced. The technique can be applied by thresholding any network map, and thus generalizes to any ICA or seed-based connectivity results. Additionally, spurious features arising above threshold in any region will be detected. For instance, a spurious effect in ventral regions of the brain can be well captured (see example in **sup. figure 4** for the group report), but would be missed by a priori seeds in the S1 and ACC. The Dice overlap measure is also complemented

*with additional measures, which are necessary to ensure a complete account of potential pitfalls during quality control (see main manuscript and **supplementary table 4**). “*

References

- Abraham, A., Pedregosa, F., Eickenberg, M., Gervais, P., Mueller, A., Kossaifi, J., Gramfort, A., Thirion, B., & Varoquaux, G. (2014). Machine learning for neuroimaging with scikit-learn. *Frontiers in Neuroinformatics*, 8, 14.
- Bright, M. G., & Murphy, K. (2015). Is fMRI “noise” really noise? Resting state nuisance regressors remove variance with network structure. *NeuroImage*, 114, 158–169.
- Celestine, M., Nadkarni, N. A., Garin, C. M., Bougacha, S., & Dhenain, M. (2020). Samba-MRI: A Library for Processing Small-Mammal Brain MRI Data in Python. *Frontiers in Neuroinformatics*, 14, 24.
- Ciric, R., Rosen, A. F. G., Erus, G., Cieslak, M., Adebimpe, A., Cook, P. A., Bassett, D. S., Davatzikos, C., Wolf, D. H., & Satterthwaite, T. D. (2018). Mitigating head motion artifact in functional connectivity MRI. *Nature Protocols*, 13(12), 2801–2826.
- Craddock, C., Sikka, S., Cheung, B., Khanuja, R., Ghosh, S. S., Yan, C., Li, Q., Lurie, D., Vogelstein, J., Burns, R., & Others. (2013). Towards automated analysis of connectomes: The configurable pipeline for the analysis of connectomes (c-pac). *Frontiers in Neuroinformatics*, 42, 10–3389.
- Diao, Y., Yin, T., Gruetter, R., & Jelescu, I. O. (2021). PIRACY: An Optimized Pipeline for Functional Connectivity Analysis in the Rat Brain. *Frontiers in Neuroscience*, 15, 602170.

- Esteban, O., Birman, D., Schaer, M., Koyejo, O. O., Poldrack, R. A., & Gorgolewski, K. J. (2017). MRIQC: Advancing the automatic prediction of image quality in MRI from unseen sites. *PLoS One*, 12(9), e0184661.
- Esteban, O., Markiewicz, C. J., Blair, R. W., Moodie, C. A., Isik, A. I., Erramuzpe, A., Kent, J. D., Goncalves, M., DuPre, E., Snyder, M., Oya, H., Ghosh, S. S., Wright, J., Durnez, J., Poldrack, R. A., & Gorgolewski, K. J. (2019). fMRIPrep: a robust preprocessing pipeline for functional MRI. *Nature Methods*, 16(1), 111–116.
- Grandjean, J., Canella, C., Anckaerts, C., Ayranci, G., Bougacha, S., Bienert, T., Buehlmann, D., Coletta, L., Gallino, D., Gass, N., Garin, C. M., Nadkarni, N. A., Hübner, N. S., Karatas, M., Komaki, Y., Kreitz, S., Mandino, F., Mechling, A. E., Sato, C., ... Gozzi, A. (2020). Common functional networks in the mouse brain revealed by multi-centre resting-state fMRI analysis. *NeuroImage*, 205, 116278.
- Grandjean, J., Desrosiers-Gregoire, G., Anckaerts, C., Angeles-Valdez, D., Ayad, F., Barrière, D. A., Blockx, I., Bortel, A., Broadwater, M., Cardoso, B. M., Célestine, M., Chavez-Negrete, J. E., Choi, S., Christiaen, E., Clavijo, P., Colon-Perez, L., Cramer, S., Daniele, T., Dempsey, E., ... Hess, A. (2023). A consensus protocol for functional connectivity analysis in the rat brain. *Nature Neuroscience*, 26(4), 673–681.
- Ioanas, H.-I., Marks, M., Zerbi, V., Yanik, M. F., & Rudin, M. (2021). An optimized registration workflow and standard geometric space for small animal brain imaging. *NeuroImage*, 241, 118386.
- Ji, J. L., Demšar, J., Fonteneau, C., Tamayo, Z., Pan, L., Kraljič, A., Matkovič, A., Purg, N., Helmer, M., Warrington, S., Winkler, A., Zerbi, V., Coalson, T. S., Glasser, M. F.,

- Harms, M. P., Sotiropoulos, S. N., Murray, J. D., Anticevic, A., & Repovš, G. (2023). QuNex-An integrative platform for reproducible neuroimaging analytics. *Frontiers in Neuroinformatics*, 17, 1104508.
- Lee, S.-H., Broadwater, M. A., Ban, W., Wang, T.-W. W., Kim, H.-J., Dumas, J. S., Vetro, R. P., Herman, M. A., Morrow, A. L., Besheer, J., Kash, T. L., Boettiger, C. A., Robinson, D. L., Crews, F. T., & Shih, Y.-Y. I. (2021). An isotropic EPI database and analytical pipelines for rat brain resting-state fMRI. *NeuroImage*, 243, 118541.
- Murphy, K., & Fox, M. D. (2017). Towards a consensus regarding global signal regression for resting state functional connectivity MRI. *NeuroImage*, 154, 169–173.
- Power, J. D., Lynch, C. J., Adeyemo, B., & Petersen, S. E. (2020). A Critical, Event-Related Appraisal of Denoising in Resting-State fMRI Studies. *Cerebral Cortex*, 30(10), 5544–5559.
- Satterthwaite, T. D., Ciric, R., Roalf, D. R., Davatzikos, C., Bassett, D. S., & Wolf, D. H. (2019). Motion artifact in studies of functional connectivity: Characteristics and mitigation strategies. *Human Brain Mapping*, 40(7), 2033–2051.
- Taylor, P. A., Glen, D. R., Reynolds, R. C., Basavaraj, A., Moraczewski, D., & Etzel, J. A. (2023). Editorial: Demonstrating quality control (QC) procedures in fMRI. *Frontiers in Neuroscience*, 17. <https://doi.org/10.3389/fnins.2023.1205928>
- Wang, S., Peterson, D. J., Gatenby, J. C., Li, W., Grabowski, T. J., & Madhyastha, T. M. (2017). Evaluation of Field Map and Nonlinear Registration Methods for Correction of Susceptibility Artifacts in Diffusion MRI. *Frontiers in Neuroinformatics*, 11, 17.

Reviewer #1 (Remarks to the Author):

The authors have addressed all of my comments in a satisfactory way.

Reviewer #2 (Remarks to the Author):

I would like to express my satisfaction with the responses and additional information provided in your rebuttal. Your efforts have greatly improved the clarity and completeness of the manuscript.

I believe that the RABIES represents one of the most commendable endeavors in creating a preprocessing platform for imaging data in the preclinical field. However, I would like to add a couple of minor points for potential improvement.

Firstly, while the requirement for BIDS inputs is understandable, it would be beneficial to include functionality for converting BRUKER files to BIDS format as part of RABIES pipeline, as BRUKER files are commonly used by a significant portion of preclinical researchers. While it might appear insignificant, for users who lack experience, this could pose a significant barrier, potentially restricting the software's utilization.

Additionally, there seems to be an issue with the inclusion/exclusion hits for changing the naming convention of input files. It would be helpful if the authors could investigate this matter further to ensure the functionality is working as intended.

Overall, I am impressed with the quality of the RABIES platform and sincerely hope that the promises for ongoing maintenance will be fulfilled.

Best regards,
Prof. Valerio Zerbi

Reviewer #2 (Remarks on code availability):

I have found no bugs on the code with only one exception that I've highlighted in my review

We thank the reviewers once again for their feedback and consideration of the manuscript. Reviewer #2 raised the following two remarks for future improvements, and we provide our answer in *italic font*.

1) Firstly, while the requirement for BIDS inputs is understandable, it would be beneficial to include functionality for converting BRUKER files to BIDS format as part of RABIES pipeline, as BRUKER files are commonly used by a significant portion of preclinical researchers. While it might appear insignificant, for users who lack experience, this could pose a significant barrier, potentially restricting the software's utilization.

Thank you for your concern regarding the accessibility of RABIES to the community. This is indeed a priority of ours in developing the software. We have opened an issue (<https://github.com/CoBrALab/RABIES/issues/368>) regarding the potential adaptation of our current package of choice (BrkRaw <https://brkraw.github.io/>) for converting Bruker files into the BIDS format. We believe, however, that tackling conversions issues brings about its own new set of technical challenges which would be best addressed by the developers of those conversion tools, rather than the RABIES developers. As such we have chosen not to implement these tools within RABIES at this time. Rather, we have chosen to direct users towards those tools in the RABIES documentation.

2) Additionally, there seems to be an issue with the inclusion/exclusion hits for changing the naming convention of input files. It would be helpful if the authors could investigate this matter further to ensure the functionality is working as intended.

We thank you for your thorough evaluation of the software. We welcome the reviewer to open an issue on our Github webpage to investigate this issue further. This will allow us to obtain and provide detailed feedback (log reports, etc...) to tackle the issue.